# Bandit-Feedback Online Multiclass Classification: Variants and Tradeoffs

**Yuval Filmus**

Faculty of Computer Science

Faculty of Mathematics

Technion, Israel

`filmus.yuval@gmail.com`

**Steve Hanneke**

Department of Computer Science

Purdue University, USA

`steve.hanneke@gmail.com`

**Idan Mehalel**

Faculty of Computer Science

Technion, Israel

`idanmehalel@gmail.com`

**Shay Moran**

Faculty of Mathematics

Faculty of Computer Science

Faculty of Data and Decision Sciences

Technion, Israel

Google research, Israel

`shaymoran1@gmail.com`

## Abstract

Consider the domain of multiclass classification within the adversarial online setting. What is the price of relying on bandit feedback as opposed to full information? To what extent can an adaptive adversary amplify the loss compared to an oblivious one? To what extent can a randomized learner reduce the loss compared to a deterministic one? We study these questions in the mistake bound model and provide nearly tight answers. We demonstrate that the optimal mistake bound under bandit feedback is at most $O(k)$ times higher than the optimal mistake bound in the full information case, where $k$ represents the number of labels. This bound is tight and provides an answer to an open question previously posed and studied by Daniely and Helbertal ['13] and by Long ['17, '20], who focused on deterministic learners. Moreover, we present nearly optimal bounds of $\tilde{\Theta}(k)$ on the gap between randomized and deterministic learners, as well as between adaptive and oblivious adversaries in the bandit feedback setting. This stands in contrast to the full information scenario, where adaptive and oblivious adversaries are equivalent, and the gap in mistake bounds between randomized and deterministic learners is a constant multiplicative factor of 2. In addition, our results imply that in some cases the optimal randomized mistake bound is approximately the square-root of its deterministic parallel. Previous results show that this is essentially the smallest it can get. Some of our results are proved via a reduction to *prediction with expert advice* under bandit feedback, a problem interesting on its own right. For this problem, we provide a randomized algorithm which is nearly optimal in some scenarios.

38th Conference on Neural Information Processing Systems (NeurIPS 2024).

# 1    Introduction

The primary focus of this work is the study of *multiclass online learning* of hypothesis classes in the *realizable* setting (in which some hypothesis perfectly labels the input). Online learning is a repeated game between a learner and an adversary. Each round $t$ in the game proceeds as follows:

 (i)  The adversary sends an instance $x_t \in \mathcal{X}$ to the learner.
 (ii)  The learner predicts $\hat{y}_t \in \mathcal{Y}$ (possibly at random).
 (iii)  The adversary provides information (feedback) to the learner.

This problem suggests a wide and appealing landscape of possible different setups, each granting or preventing various resources from the learner or the adversary which generates the input. The resources we discuss are the *information* provided by the adversary, the *adaptivity* of the adversary, and the *randomness* of the learner. *Full-information* feedback means that the learner learns the correct label $y_t$ at the end of every round. *Bandit* feedback means that the learner only receives an indication whether $\hat{y}_t = y_t$ or not. If the learner predicts $\hat{y}_t$ deterministically for every $t$, then the learner is *deterministic*. See Section A for formal and self-contained definitions.

We are interested in how the optimal mistake bound is affected by granting or preventing each of those resources. Below, we discuss these questions and present our results.

## 1.1    Main questions and results

We assume that a concept class $\mathcal{H} \in \mathcal{Y}^{\mathcal{X}}$ is given, where $\mathcal{X}$ is a *domain* of instances and $\mathcal{Y}$ is a *label set*. Unless stated otherwise, we restrict the learning scenario to the realizable case, in which the input provided by the adversary is consistent with a concept from $\mathcal{H}$.

In order to state the results, we define some notation. For a concept class $\mathcal{H}$, let $\mathsf{opt}_{\mathrm{full}}^{\mathrm{rand}}(\mathcal{H})$ denote the optimal mistake bound for $\mathcal{H}$ achievable with full-information feedback. Let $\mathsf{opt}_{\mathrm{full}}^{\mathrm{det}}(\mathcal{H})$ denote the same as $\mathsf{opt}_{\mathrm{full}}^{\mathrm{rand}}(\mathcal{H})$, with the additional restriction that the learner is deterministic. In the bandit feedback model, define $\mathsf{opt}_{\mathrm{bandit}}^{\mathrm{det}}(\mathcal{H})$ analogously to $\mathsf{opt}_{\mathrm{full}}^{\mathrm{det}}(\mathcal{H})$. When the learner is allowed to be randomized, the mistake bound in the bandit feedback model might change if the game is played against an oblivious or adaptive adversary (we define these types of adversaries below).[1] Therefore, we define the notations $\mathsf{opt}_{\mathrm{bandit}}^{\mathrm{obl}}(\mathcal{H})$ and $\mathsf{opt}_{\mathrm{bandit}}^{\mathrm{adap}}(\mathcal{H})$ to denote the optimal mistake bounds of a randomized learner which receives bandit feedback on its predictions, when the adversary is oblivious or adaptive, respectively. Unless stated otherwise, $O$ and $\Omega$ notations hide universal constants that do not depend on any parameter of the problem.

### 1.1.1    Information

In the full-information feedback model, the learner learns the correct label at the end of each round, regardless of its prediction. In the bandit feedback model, significantly less information is provided at the end of each round: the adversary only reveals whether the learner's prediction was correct or incorrect. This raises the following natural question studied e.g. by Auer and Long [1999], Daniely and Helbertal [2013], Daniely, Sabato, Ben-David, and Shalev-Shwartz [2015], Long [2020].

> What is the price the learner pays for receiving only bandit feedback?

The answer for this question is known for deterministic learners. Auer and Long [1999] proved the upper bound
$$\mathsf{opt}_{\mathrm{bandit}}^{\mathrm{det}}(\mathcal{H}) = O(\mathsf{opt}_{\mathrm{full}}^{\mathrm{det}}(\mathcal{H}) \cdot |\mathcal{Y}| \log |\mathcal{Y}|) \tag{1}$$
for every concept class $\mathcal{H}$. A matching lower bound was given in [Long, 2020, Geneson, 2021], which showed that for every natural $k \geq 2$ there exists a concept class $\mathcal{H}$ with label set of size

---

[1]In the full-information model there is no essential difference between oblivious and adaptive adversaries (as long as the learner uses fresh randomness in every round) [Cesa-Bianchi and Lugosi, 2006, Lemma 4.1]. Essentially, the reason is that in the full-information model the feedback is deterministic, even if the learner is randomized.

$|\mathcal{Y}| = k$ such that

$$\mathsf{opt}_{\mathrm{bandit}}^{\mathrm{det}}(\mathcal{H}) = \Omega(\mathsf{opt}_{\mathrm{full}}^{\mathrm{det}}(\mathcal{H}) \cdot |\mathcal{Y}| \log |\mathcal{Y}|). \tag{2}$$

They also find tight guarantees on the constants hidden in the $O, \Omega$ notations.

Finding an analogous result for randomized learners was raised as an open problem in [Daniely and Helbertal, 2013].[2] We solve this open problem by proving the upper bound stated in the following theorem.

**Theorem 1.1** (Full-information vs. Bandit-feedback). *For every concept class $\mathcal{H}$ it holds that*

$$\mathsf{opt}_{\mathrm{bandit}}^{\mathrm{adap}}(\mathcal{H}) = O(\mathsf{opt}_{\mathrm{full}}^{\mathrm{rand}}(\mathcal{H}) \cdot |\mathcal{Y}|).$$

*Furthermore, for every natural $k \geq 2$ there exists a concept class $\mathcal{H}$ with $|\mathcal{Y}| = k$ such that*

$$\mathsf{opt}_{\mathrm{bandit}}^{\mathrm{obl}}(\mathcal{H}) = \Omega(\mathsf{opt}_{\mathrm{full}}^{\mathrm{rand}}(\mathcal{H}) \cdot |\mathcal{Y}|).$$

The lower bound was noted e.g. in [Daniely and Helbertal, 2013]. We complement it by proving an upper bound that holds for all classes. We prove Theorem 1.1 in Section D. Note that $\mathsf{opt}_{\mathrm{full}}^{\mathrm{rand}}(\mathcal{H})$ is precisely characterized by a combinatorial parameter of $\mathcal{H}$ called the *randomized Littlestone dimension* [Filmus, Hanneke, Mehalel, and Moran, 2023] (and characterized up to a multiplicative factor of 2 by the standard *Littlestone dimension* [Littlestone, 1988]).

The techniques developed to prove this upper bound include new bounds for *prediction with expert advice* with bandit feedback, which is the main technical contribution of this work. These bounds are presented in Section 1.2, and a technical overview of the proof can be found in Section 2.

**A generalization to the agnostic setting.** We also generalize results in the spirit of Theorem 1.1 to the agnostic case, in which the best hypothesis in class is inconsistent with the input in $r^\star$ many rounds. Our results do not require that $r^\star$ (or some bound $r \geq r^\star$ on it) is given to the learner. However, throughout most of the paper we assume the stronger *r-realizability* assumption, in which an upper bound $r \geq r^\star$ is given to the learner. That is, under $r$-realizability assumption the learner knows in advance that some hypothesis will be inconsistent with the feedback in *at most $r$ many* rounds. We explain in Section G how to remove this assumption using a standard "doubling trick". Bounds for various setups in the agnostic setting are summarized in Table 1.

In the agnostic setting, learning algorithms are often measured by their expected *regret* (which is the mistake bound minus the number of mistakes made by the best hypothesis). While in this work we measure algorithms by their mistake bound, note that a mistake bound of an algorithm is always at least as large as its regret. Therefore, since our bounds demonstrate no dependence on the number of rounds $T$, in some cases they provide improvements over the known regret bound $\tilde{O}\left(\sqrt{T|\mathcal{Y}|\mathsf{opt}_{\mathrm{full}}^{\mathrm{det}}(\mathcal{H})}\right)$ of [Daniely and Helbertal, 2013]. Specifically, when $r^\star = O(\mathsf{opt}_{\mathrm{full}}^{\mathrm{det}}(\mathcal{H}))$, our results imply a regret bound of $O(|\mathcal{Y}|\mathsf{opt}_{\mathrm{full}}^{\mathrm{det}}(\mathcal{H}))$, for all $T$.

### 1.1.2 Adaptivity

In Section 1.1.1 we showed that randomness is necessary for obtaining optimal bounds on the cost of bandit feedback. However, within the setup of a randomized learner which receives bandit feedback, there are (at least) two types of adversaries to consider: an oblivious adversary which must decide on the entire input in advance, and an adaptive adversary that can decide on the input on the fly. This raises the following natural question.

> Within the bandit feedback model, what is the price the learner pays for playing against an adaptive adversary?

We solve this question up to logarithmic factors. However, our nearly tight lower bound uses *pattern classes*, which are a generalization of concept classes. In more detail, a pattern class is a set of

---

[2]The setting considered in [Daniely and Helbertal, 2013], is, however, more general than the setting we consider. They consider an adversary which can decide on a few correct labels in each round, and not only one. Generalizing this work to their multilabel setting is an interesting direction for future work.

| Learner \ Adversary | Oblivious | Adaptive |
|---|---|---|
| Randomized | $O\big(|\mathcal{Y}|(\mathsf{opt}_{\mathrm{full}}^{\mathrm{det}}(\mathcal{H}) + r^\star)\big)$ 
 $\Omega\big(|\mathcal{Y}| \cdot \mathsf{opt}_{\mathrm{full}}^{\mathrm{det}}(\mathcal{H}) + r^\star\big)$ | $\Theta\big(|\mathcal{Y}|(\mathsf{opt}_{\mathrm{full}}^{\mathrm{det}}(\mathcal{H}) + r^\star)\big)$ |
| Deterministic | $O\big(|\mathcal{Y}| \log |\mathcal{Y}|(\mathsf{opt}_{\mathrm{full}}^{\mathrm{det}}(\mathcal{H}) + r^\star)\big)$ 
 $\Omega\big(|\mathcal{Y}|(\mathsf{opt}_{\mathrm{full}}^{\mathrm{det}}(\mathcal{H}) \log |\mathcal{Y}| + r^\star)\big)$ | |

Table 1: Worst-case mistake bounds for learning concept classes with bandit feedback in the agnostic setting, in which the best concept in class is inconsistent with the feedback in $r^\star$ many rounds. No prior knowledge on $r^\star$ is required. The bounds for the randomized setup are obtained by applying Theorem D.3 to the upper bound in Theorem D.1 and to the lower bounds in Lemma C.11 and in [Daniely and Helbertal, 2013]. The bounds for the deterministic setup are obtained by applying the same theorem to the upper bound of Auer and Long [1999], and to the lower bounds of Lemma C.5 and Long [2020]. The same (up to constant) deterministic bounds were proved independently in [Geneson and Tang, 2024] (however, their bounds are in terms of a given bound $r \geq r^\star$).

*patterns* $p \in (\mathcal{X} \times \mathcal{Y})^\star$ which is downwards closed (i.e. closed under sub-patterns). Proving the lower bound in the theorem below using only concept classes or showing that this is not possible is an interesting and main problem left open by this work.

**Theorem 1.2** (Oblivious vs. Adaptive Adversaries). *For every concept class $\mathcal{H}$ it holds that*

$$\mathsf{opt}_{\mathrm{bandit}}^{\mathrm{adap}}(\mathcal{H}) = O(\mathsf{opt}_{\mathrm{bandit}}^{\mathrm{obl}}(\mathcal{H}) \cdot |\mathcal{Y}| \log |\mathcal{Y}|).$$

*Furthermore, for every natural $k \geq 2$ there exists a pattern class $\mathcal{P}$ with $|\mathcal{Y}| = k$ and so that*

$$\mathsf{opt}_{\mathrm{bandit}}^{\mathrm{adap}}(\mathcal{P}) = \Omega(\mathsf{opt}_{\mathrm{bandit}}^{\mathrm{obl}}(\mathcal{P}) \cdot |\mathcal{Y}|).$$

The upper bound is an immediate corollary of the upper bound (1). We prove the lower bound in Section E. The proof idea of the lower bound is to consider the classic adversarial $|\mathcal{Y}|$-armed bandit problem of Auer, Cesa-Bianchi, Freund, and Schapire [2002] with an $r$-realizability assumption. This setting can be simulated using a pattern class, but not using a concept class. For this problem, an adaptive adversary can force a mistake bound of $\Omega(r \cdot |\mathcal{Y}|)$, while the best an oblivious adversary can do is $O(r + |\mathcal{Y}|)$.

**Remark 1.3** (Concept classes vs. Pattern classes). *Pattern classes are a more expressive generalization of concept classes (see a formal definition in Section A). Most desirably, we would like to prove upper bounds for all pattern classes, accompanied with tight lower bounds that hold for hard concept classes. We manage to do so in all of our results (for the sake of simplicity, we omitted it from the theorem statements), except for Theorem 1.2, in which the upper bound does hold for all pattern classes, but the lower bound holds only for hard pattern classes.*

### 1.1.3 Randomness

In Section 1.1.2 we discussed two different adversarial models within the setup of randomized learners. However, what happens if the learner cannot use randomness? A folklore result on the full-information feedback model states that for every concept class $\mathcal{H}$ with $|\mathcal{Y}| = 2$ it holds that

$$\mathsf{opt}_{\mathrm{full}}^{\mathrm{rand}}(\mathcal{H}) \leq \mathsf{opt}_{\mathrm{full}}^{\mathrm{det}}(\mathcal{H}) \leq 2 \cdot \mathsf{opt}_{\mathrm{full}}^{\mathrm{rand}}(\mathcal{H}), \tag{3}$$

and that there are classes attaining each equality. Extending this to $|\mathcal{Y}| > 2$ is straightforward by noting the following randomized-to-deterministic full-information algorithm conversion: If $A$ is a randomized algorithm, then the mistake bound of the deterministic algorithm $A'$ who always predicts the most probable prediction of $A$ is at most twice the mistake bound of $A$. Indeed, whenever $A'$ makes a mistake, $A$ makes a mistake with probability at least $1/2$. This raises the following natural question, also asked in [Daniely and Helbertal, 2013].

> Within the bandit feedback model, what is the price the learner pays for not using randomness?

We resolve this question up to logarithmic factors.

**Theorem 1.4** (Randomized vs. Deterministic). *For every concept class $\mathcal{H}$ it holds that*

$$\mathsf{opt}_{\mathrm{bandit}}^{\mathrm{det}}(\mathcal{H}) = O(\mathsf{opt}_{\mathrm{bandit}}^{\mathrm{obl}}(\mathcal{H}) \cdot |\mathcal{Y}| \log |\mathcal{Y}|).$$

*Furthermore, for every natural $k \geq 2$ there exists a concept class $\mathcal{H}$ with $|\mathcal{Y}| = k$ such that*

$$\mathsf{opt}_{\mathrm{bandit}}^{\mathrm{det}}(\mathcal{H}) = \Omega(\mathsf{opt}_{\mathrm{bandit}}^{\mathrm{adap}}(\mathcal{H}) \cdot |\mathcal{Y}|).$$

The upper bound is an immediate corollary of the upper bound (1). The proof idea of the lower bound is to construct a class which is hard for a deterministic learner, but becomes easy if the learner may use randomness. Concretely, we prove the following result from which Theorem 1.4 follows, in Section F.

**Theorem 1.5.** *For every $d \geq 1$ and $k \geq 2$ there exists a concept class $\mathcal{H} \subset \mathcal{Y}^{\mathcal{X}}$ with $\mathcal{Y} = \{0, 1, \ldots, k\}$ such that*

1. $\mathsf{opt}_{\mathrm{full}}^{\mathrm{det}}(\mathcal{H}) = d + 1.$

2. $\mathsf{opt}_{\mathrm{bandit}}^{\mathrm{det}}(\mathcal{H}) = \Theta(d \cdot k).$

3. $\mathsf{opt}_{\mathrm{bandit}}^{\mathrm{adap}}(\mathcal{H}) = \Theta(d + k).$

The lower bound in Theorem 1.4 has a significant consequence on the problem of finding a combinatorial dimension that quantifies the optimal mistake bound of randomized learners in the bandit feedback model. In the full-information model, the optimal deterministic mistake bound, which is captured precisely by the combinatorial *Littlestone dimension* [Littlestone, 1988], quantifies the optimal mistake bound of randomized learners as well, as demonstrated in (3). In [Daniely, Sabato, Ben-David, and Shalev-Shwartz, 2015], a new combinatorial dimension, coined the *Bandit-Littlestone dimension*, is introduced and proved to capture the exact optimal mistake bound of deterministic learners within the bandit feedback model. In [Daniely and Helbertal, 2013], some hope is expressed for this dimension to quantify the mistake bound of randomized learners as well, similarly to the case of full-information feedback. However, our lower bound shows that this is not the case. As we show in Section F, the classes used in the lower bound may be chosen such that $\mathsf{opt}_{\mathrm{bandit}}^{\mathrm{adap}}(\mathcal{H}) = O(|\mathcal{Y}|)$, and thus $\mathsf{opt}_{\mathrm{bandit}}^{\mathrm{adap}}(\mathcal{H})$ is only roughly $\sqrt{\mathsf{opt}_{\mathrm{bandit}}^{\mathrm{det}}(\mathcal{H})}$. On the other hand, the upper bound in Theorem 1.4 together with the bound $\mathsf{opt}_{\mathrm{bandit}}^{\mathrm{obl}}(\mathcal{H}) \geq \frac{|\mathcal{Y}|-1}{2}$ (e.g. by Daniely and Helbertal [2013]) shows that $\sqrt{\mathsf{opt}_{\mathrm{bandit}}^{\mathrm{det}}(\mathcal{H})}$ is roughly the smallest that $\mathsf{opt}_{\mathrm{bandit}}^{\mathrm{obl}}(\mathcal{H})$ can get.

### 1.2 Bounds for *prediction with expert advice*

A main technical result proved in this work, which is interesting in its own right, is a nearly optimal randomized mistake bound for the problem of *prediction with expert advice* in the $r$-realizable setting. In this problem, $n$ experts make deterministic predictions in every round, and it is promised that the best expert is inconsistent with the feedback for at most $r$ many times throughout the entire game. As in Section 1.1.1, the knowledge of $r$ is actually not required, and $r$ can be replaced with the actual number of inconsistencies of the best expert, $r^\star$. In Section G, we explain how to remove the assumption that $r$ is given to the learner.

The learner should aggregate the experts' predictions to make their own (possibly randomized) predictions, while minimizing the expected number of mistakes made. This problem was extensively studied in the binary ($|\mathcal{Y}| = 2$) setting, starting with the seminal works of Vovk [1990], Littlestone and Warmuth [1994] who showed that the optimal mistake bound when $|\mathcal{Y}| = 2$ is $\Theta(\log n + r)$ for both randomized and deterministic learners. Later, Cesa-Bianchi, Freund, Helmbold, and Warmuth [1996] proved fine-grained bounds for deterministic learners. Cesa-Bianchi, Freund, Haussler, Helmbold, Schapire, and Warmuth [1997], Abernethy, Langford, and Warmuth [2006], Filmus,

| Adversary / Learner | Oblivious | Adaptive |
|---|---|---|
| Randomized | $O(k(\log_k n + r^\star))$ $\Omega(k \log_k n + r^\star)$ | $\Theta(k(\log_k n + r^\star))$ |
| Deterministic | $\Theta(k(\log(n/k) + r^\star + 1))$ | |

Table 2: Mistake bounds for *prediction with expert advice*. The size of the label set is $k \geq 2$, there are $n \geq k$ experts, and the best expert is inconsistent with the feedback for $r^\star$ many times. No prior knowledge on $r^\star$ is required. The randomized bounds are due to Theorem C.7 and Lemmas C.10 and C.11. The deterministic bounds are stated in Theorem C.1.

Hanneke, Mehalel, and Moran [2023] further refined and improved randomized mistake bounds. Mukherjee and Schapire [2010] studied a variation where the experts are randomized and the learner is deterministic. Similarly to this paper, Brânzei and Peres [2019] studied the multiclass scenario, but with full-information-feedback, which is substantially easier to the learner.

In this work, we consider this problem in the bandit feedback model. Most previous works on prediction with expert advice under bandit feedback studied the best achievable *regret*, obtaining results that depend on the number of rounds $T$, a dependence from which we seek to avoid in this work (see Section 3.4 for more details).

The main tool used to prove Theorem 1.1 is the following optimal (up to constant factors) bound on $\mathsf{opt}_{\mathrm{bandit}}^{\mathrm{adap}}(n, k, r)$, which is the optimal mistake bound achievable by a randomized learner which plays against an adaptive adversary that provides bandit feedback, when there are $k \geq 2$ many labels, $n \geq k$ many experts, and the best expert is inconsistent with the feedback for at most $r \geq 0$ many times.

**Theorem 1.6.** *For every $n \geq k \geq 2$ and $r \geq 0$ it holds that*

$$\mathsf{opt}_{\mathrm{bandit}}^{\mathrm{adap}}(n, k, r) = \Theta(k(\log_k n + r)).$$

This bound generalizes the result $\mathsf{opt}_{\mathrm{bandit}}^{\mathrm{adap}}(n, 2, r) = \Theta(\log n + r)$ mentioned above.[3] In the case where the adversary is oblivious, we prove the inferior lower bound

$$\mathsf{opt}_{\mathrm{bandit}}^{\mathrm{obl}}(n, k, r) = \Omega(k \log_k n + r)$$

which is tight as long as $r = O(\log_k n)$. Proving tight bounds against an oblivious adversary for all values of $r$ remains open.

We also consider this problem in the deterministic (learner) setting, for the sake of completeness (we do not use the deterministic bound to prove any other results). We prove all bounds in Section C, and summarize them in Table 2. Similarly to the results in Section 1.1.1, since our mistake bounds demonstrate no dependence on the number of rounds $T$, they improve over the known regret bound $O(\sqrt{Tk \log n})$ of Auer, Cesa-Bianchi, Freund, and Schapire [2002] whenever $r = O(\log n)$.

## 2 Technical overview

In this section, we explain the idea behind the proof of the upper bound $\mathsf{opt}_{\mathrm{bandit}}^{\mathrm{adap}}(\mathcal{H}) = O(\mathsf{opt}_{\mathrm{full}}^{\mathrm{rand}}(\mathcal{H}) \cdot |\mathcal{Y}|)$ in Theorem 1.1, which is our main technical contribution.

There are two main ingredients used in the proof of Theorem 1.1:

1. A reduction from the problem of learning a concept class $\mathcal{H}$ to an instance of *prediction with expert advice* with $|\mathcal{Y}|^{\mathsf{opt}_{\mathrm{full}}^{\mathrm{det}}(\mathcal{H})}$ many experts, $|\mathcal{Y}|$ many labels, and where the best expert is inconsistent with the feedback for at most $\mathsf{opt}_{\mathrm{full}}^{\mathrm{det}}(\mathcal{H})$ many rounds.

---

[3]When there are only 2 labels, full-information and bandit feedback are the same.

2. The upper bound for *prediction with expert advice* stated in Theorem 1.6.

Indeed, having both items, we take the upper bound of item (2) with the parameters specified in item (1), obtaining

$$\mathsf{opt}_{\mathrm{bandit}}^{\mathrm{adap}}\left(|\mathcal{Y}|^{\mathsf{opt}_{\mathrm{full}}^{\mathrm{det}}(\mathcal{H})}, |\mathcal{Y}|, \mathsf{opt}_{\mathrm{full}}^{\mathrm{det}}(\mathcal{H})\right) = O\left(|\mathcal{Y}| \cdot \mathsf{opt}_{\mathrm{full}}^{\mathrm{det}}(\mathcal{H})\right).$$

By item (1), this bound holds also for the problem of learning the concept class $\mathcal{H}$. Since $\mathsf{opt}_{\mathrm{full}}^{\mathrm{det}}(\mathcal{H}) \leq 2\mathsf{opt}_{\mathrm{full}}^{\mathrm{rand}}(\mathcal{H})$, we can replace $\mathsf{opt}_{\mathrm{full}}^{\mathrm{det}}(\mathcal{H})$ with $\mathsf{opt}_{\mathrm{full}}^{\mathrm{rand}}(\mathcal{H})$. It remains to sketch the ideas behind the proofs of items (1) and (2), which we do in the following subsections. Of the two items, the proof of Item (2) is the main technical novelty.

## 2.1 Proof idea of Item (1)

In the problem of *prediction with expert advice*, the expert predictions are generated in a completely adversarial fashion. That is, no assumptions are made on the way those predictions are generated. Therefore, any upper bound for *prediction with expert advice* holds in particular for the case where the expert predictions are determined by an algorithm chosen by the learner. We can exploit this property to reduce the problem of learning a concept class $\mathcal{H}$ to an instance of *prediction with expert advice* with $|\mathcal{Y}|^{\mathsf{opt}_{\mathrm{full}}^{\mathrm{det}}(\mathcal{H})}$ many experts, $|\mathcal{Y}|$ many labels, and where the best expert is inconsistent with the feedback for at most $\mathsf{opt}_{\mathrm{full}}^{\mathrm{det}}(\mathcal{H})$ many rounds. The idea, described below, is inspired by Long [2020], Hanneke, Livni, and Moran [2021].

Let $A$ be an optimal deterministic algorithm for learning $\mathcal{H}$ given full-information feedback. We run in parallel several copies of $A$ arranged in tree form. These copies will function as the experts fed into an optimal algorithm for *prediction with expert advice*.

Initially, there is a single copy of $A$. At every round, for each copy of $A$, if its prediction is consistent with the bandit feedback, we do nothing. Otherwise, if the copy is at depth $D < \mathsf{opt}_{\mathrm{full}}^{\mathrm{det}}(\mathcal{H})$, we split it into $|\mathcal{Y}|$ different copies at depth $D + 1$, each "guessing" a different full-information feedback for the problematic example; if the copy is at depth $D$, we do nothing.

Since the tree has depth at most $\mathsf{opt}_{\mathrm{full}}^{\mathrm{det}}(\mathcal{H})$, there are at most $|\mathcal{Y}|^{\mathsf{opt}_{\mathrm{full}}^{\mathrm{det}}(\mathcal{H})}$ many copies of $A$. The copy of $A$ which always guessed correctly corresponds to an expert whose predictions are inconsistent with the feedback for at most $\mathsf{opt}_{\mathrm{full}}^{\mathrm{det}}(\mathcal{H})$ many rounds.

A formal statement (and proof) of this reduction can be found in Proposition D.2.

## 2.2 Proof idea of Item (2)

In the context of classification problems, the realizable case in which some concept from the learned class accurately explains the correct classification is often simpler than the agnostic case. Therefore, for the sake of understanding the proof of the upper bound on $\mathsf{opt}_{\mathrm{bandit}}^{\mathrm{adap}}(n, k, r)$ stated in Theorem 1.6, we first outline a proof for the upper bound on $\mathsf{opt}_{\mathrm{bandit}}^{\mathrm{adap}}(n, k) := \mathsf{opt}_{\mathrm{bandit}}^{\mathrm{adap}}(n, k, 0)$. This proof has the same flavor of the proof for general $r$, but is simpler. We then explain how to adapt the proof for general $r$.

When $r = 0$, all *living* experts (that is, experts which have been consistent with the feedback so far) are identical: every expert which is inconsistent with the feedback in some round is immediately eliminated, and its predictions need not be taken into account any longer. By applying the law of total expectation, the optimal mistake bound can thus be described by the following optimization problem. Fix $k \geq 2$, and for every $n \geq 1$, let $V(n) = \mathsf{opt}_{\mathrm{bandit}}^{\mathrm{adap}}(n, k)$. We have

$$V(n) = \max_{\vec{\alpha}} \min_{\pi} \max_{y \in \mathcal{Y}} \left[ \pi_y V(\alpha_y n) + \sum_{y' \neq y} \pi_{y'} (1 + V((1 - \alpha_{y'})n)) \right], \tag{4}$$

where $\vec{\alpha} \in [0, 1]^k$ is a $k$-ary vector whose $y$'th entry specifies the fraction of living experts predicting $y$; $\pi$ is the distribution used by the learner to draw the prediction; and $y \in \mathcal{Y}$ is the correct label chosen by the adversary.

Since all living experts are identical, the natural intuition suggests that the optimal choice of $\vec{\alpha}$ is to let every entry in it be $1/k$, in which case the optimal choice of $\pi$ would be the uniform distribution. In this case, it does not matter which $y$ the adversary chooses as the correct label. Applying this intuition to (4) results in the recurrence relation

$$V(n) \leq V(n/k)/k + (1 - 1/k)(1 + V((1 - 1/k)n)). \tag{5}$$

Solving this recurrence relation gives $V(n) \leq k \log_{b(k)} n$, where $b(k) = \frac{k^k}{(k-1)^{k-1}}$, which implies the statement in Theorem 1.6 since $b(k) \geq k$. It remains to show that the intuition that led us from (4) to (5) is indeed correct. A natural approach would be to directly prove that $V(n)$, as defined in (4), satisfies $V(n) \leq k \log_{b(k)} n$ by induction on $n$. However, note that (4) contains $k$ different recursive calls, so an inductive proof seems complicated. To overcome this, we use minimax duality, which significantly simplifies (4), as we now explain.

Observe that once $\vec{\alpha}$ is fixed, each round of the game is a zero-sum game between two randomized parties.[4] Therefore, we can apply von Neumann's minimax theorem and obtain a *dual* game which is equivalent to the *primal* (original) game in terms of its optimal mistake bound.[5] In the dual game, the adversary first chooses a distribution over the labels from which the correct label is drawn, and then the learner chooses its prediction. When the dual game is considered, the definition of $V(n)$ from (4) becomes

$$V(n) = \max_{\vec{\alpha}, \pi} \min_{y \in \mathcal{Y}} [\pi_y V(\alpha_y n) + (1 - \pi_y)(1 + V((1 - \alpha_y)n))]. \tag{6}$$

The notation is the same as in (4), except that now $\pi$ is the distribution used by the adversary to draw the correct label, and $y \in \mathcal{Y}$ is the learner's prediction. Eq. (6) contains only two recursive calls, making an inductive proof much easier (but still quite technical).

We now explain how to adapt this approach to work for general $r$, when $r$ is given. In Section G, we explain how to remove the assumption that $r$ is given to the learner. When $r > 0$, there is an inherent difference between the experts: different experts have been inconsistent with the feedback for a different number of rounds, and so can afford a different number of future inconsistencies. Therefore, we need some mechanism that differentiates between experts with different inconsistency budgets.

Inspired by *weighted prediction* techniques (See [Cesa-Bianchi and Lugosi, 2006, Section 2.1] and bibliographic remarks of Section 2), we rely on the fact that experts with higher budgets have higher *potential* to damage the learner, so we choose an *expert potential function* that matches the potential of an expert to damage the learner. As such, the potential should be an increasing function of the budget. Choosing the potential as a function of the budget is a known method (see, e.g., [Cesa-Bianchi and Lugosi, 2006, Corollary 2.4]. We use exponential potentials and give an expert with budget $i$ (corresponding to $i$ more allowed inconsistencies) a potential of $k^{2i}$. We now employ roughly the same argument outlined above for the realizable case. The main difference is that $V(\cdot)$ depends on the total potential rather than on the number of living experts. As a result, the initial number of living experts $n$ is replaced with the initial total potential $n \cdot k^{2r}$, which gives

$$V(n \cdot k^{2r}) \leq k \log_k(n \cdot k^{2r}) = k \log_k n + 2kr,$$

as stated in Theorem 1.6.

## 3   Related work

We outline connections between previous work to the problems in the focus of this work: the role of various resources in realizable multiclass online learning of concept classes, and *prediction with expert advice* with bandit feedback.

---

[4]The adversary described in (4) is deterministic, but we can allow it to be randomized without changing the value of $V(n)$.

[5]Similar usage of minimax duality appears in previous work, for example [Abernethy, Agarwal, Bartlett, and Rakhlin, 2009].

## 3.1 Information

The role of information in learning concept classes was previously studied in [Auer and Long, 1999, Daniely, Sabato, Ben-David, and Shalev-Shwartz, 2015, Daniely and Helbertal, 2013, Long, 2020]. To the best of our knowledge, [Auer and Long, 1999] was the first to show that $\mathsf{opt}^{\mathrm{det}}_{\mathrm{bandit}}(\mathcal{H}) = O(\mathsf{opt}^{\mathrm{det}}_{\mathrm{full}}(\mathcal{H})|\mathcal{Y}| \log |\mathcal{Y}|)$ for every class $\mathcal{H}$. Long [2020], Geneson [2021] improved the constant in the upper bound of Auer and Long [1999], and showed that it is in fact tight up to a $1 + o(1)$ factor by finding a sequence of concept classes demonstrating a matching separation between $\mathsf{opt}^{\mathrm{det}}_{\mathrm{bandit}}(\mathcal{H})$ and $\mathsf{opt}^{\mathrm{det}}_{\mathrm{full}}(\mathcal{H})$. Our work proves an analogous result for randomized learners (Theorem 1.1), with the exception that we do not identify the exact leading constant in the optimal mistake bound.

The agnostic case was studied in [Daniely and Helbertal, 2013], which proved the upper bound $\tilde{O}\left(\sqrt{T|\mathcal{Y}|\mathsf{opt}^{\mathrm{det}}_{\mathrm{full}}(\mathcal{H})}\right)$ on the optimal regret, where $T$ is the horizon, and showed that the upper bound is best-possible up to logarithmic factors. The PAC learning setting was studied in [Daniely, Sabato, Ben-David, and Shalev-Shwartz, 2015], which showed that the price of bandit feedback is $\tilde{O}(|\mathcal{Y}|)$ in this setting as well.

## 3.2 Adaptivity

In the setting of full-information feedback, adaptive and oblivious adversaries are essentially equivalent [Cesa-Bianchi and Lugosi, 2006, Lemma 4.1]. Indeed, full-information feedback, in its essence, implies that the feedback never depends on the prediction drawn by the learner in a specific execution of the game. In the bandit feedback setting, to the best of our knowledge, the existing literature on adaptive adversaries focuses on the agnostic setting and analyze different notions of *regret*. Notable examples are [Merhav, Ordentlich, Seroussi, and Weinberger, 2002, Farias and Megiddo, 2006, Arora, Dekel, and Tewari, 2012].

## 3.3 Randomness

In the full-information feedback model, the *Littlestone dimension* [Littlestone, 1988, Daniely, Sabato, Ben-David, and Shalev-Shwartz, 2015] captures $\mathsf{opt}^{\mathrm{det}}_{\mathrm{full}}(\mathcal{H})$ precisely, and $\mathsf{opt}^{\mathrm{rand}}_{\mathrm{full}}(\mathcal{H})$ up to a multiplicative factor of 2. The paper [Daniely et al., 2015] suggests a new combinatorial dimension, the *Bandit Littlestone dimension*, and proves that it captures $\mathsf{opt}^{\mathrm{det}}_{\mathrm{bandit}}(\mathcal{H})$ precisely. The paper [Raman, Raman, Subedi, Mehalel, and Tewari, 2023] shows that the Bandit Littlestone dimension qualitatively characterizes learnability also in the agnostic and randomized setting, even when $|\mathcal{Y}| = \infty$.

The papers [Daniely and Helbertal, 2013, Raman et al., 2023] ask whether the Bandit Littlestone dimension is a good quantitative proxy for $\mathsf{opt}^{\mathrm{obl}}_{\mathrm{bandit}}(\mathcal{H})$. Theorem 1.4 (and Theorem F.1 in a more detailed version) shows that this dimension is far from quantifying even $\mathsf{opt}^{\mathrm{adap}}_{\mathrm{bandit}}(\mathcal{H})$.

## 3.4 Prediction with expert advice

The problem of *prediction with expert advice* in the $r$-realizable setting was introduced in [Vovk, 1990, Littlestone and Warmuth, 1994] for the binary case ($|\mathcal{Y}| = 2$). Since then, this problem has been well-studied, with papers spanning the past 30 years [Cesa-Bianchi, Freund, Helmbold, and Warmuth, 1996, Cesa-Bianchi, Freund, Haussler, Helmbold, Schapire, and Warmuth, 1997, Abernethy, Langford, and Warmuth, 2006, Mukherjee and Schapire, 2010, Filmus, Hanneke, Mehalel, and Moran, 2023]. The full-information setting for $|\mathcal{Y}| > 2$ and small $r$ was studied in [Brânzei and Peres, 2019].

The bandit feedback variation of the problem was introduced in [Auer, Cesa-Bianchi, Freund, and Schapire, 2002], in a more general version that considers *rewards* instead of *losses* (or *mistakes*). However, there is a major difference between how they define $r$-realizability and how we define it (in the setting of an adaptive adversary). In their definition, there must be an expert which makes at most $r$ mistakes, whereas we only assume that there must be an expert whose predictions are inconsistent with the bandit feedback at most $r$ times.

When the adversary is oblivious, both notions of $r$-realizability coincide (see Remark A.1 for more details). Therefore, it is possible that their techniques can be used to obtain tight bounds for the

oblivious setting for all values of $r$; our bounds for the adaptive setting are tight in the oblivious setting only for small $r$ values (see Theorem C.12).

Unfortunately, to the best of our knowledge, when converting rewards to losses and applying the bounds of Auer, Cesa-Bianchi, Freund, and Schapire [2002], a dependence on the number of rounds $T$ emerges (see for example [Bubeck and Cesa-Bianchi, 2012, Theorem 3.1] and [Cesa-Bianchi and Lugosi, 2006, Theorem 6.10]), which is undesirable when studying the mistake bound model (rather than the regret model). In this work we are mainly interested in the experts setting as a means for proving Theorem 1.1, and our proof specifically requires the adaptive setting. We leave for future work the question of finding mistake bounds against oblivious adversaries which are tight for large values of $r$.

## 4 Open questions and future work

Our work leaves some interesting open questions and directions for future work.

### 4.1 Open questions

**The price of adaptivity for concept classes.**    Our proof of Theorem E.1 uses pattern classes. Can we prove it using only concept classes, similarly to Theorem F.1? If not, what is the price of adaptivity when learning concept classes?

**The exact role of randomness.**    There is a $\Theta(\log k)$ gap between the lower and upper bounds in Theorem 1.4. What is the correct worst case price of not using randomness?

**The agnostic setting with oblivious adversaries.**    Our mistake bounds in the agnostic setting are not tight for large $r^\star$ when the adversary is oblivious, both for the problem of learning a concept class (Table 1), and for *prediction with expert advice* (Table 2). It would be interesting to obtain bounds that are tight for large $r^\star$ against an oblivious adversary for both problems. One possible approach towards proving such bounds for *prediction with expert advice* is to use the techniques of Auer, Cesa-Bianchi, Freund, and Schapire [2002], and specifically the Exp4 algorithm.

**A natural algorithm for the experts setting.**    Our randomized algorithm for *prediction with expert advice* is optimal, but not very natural nor efficient, as it relies on the calculation of minimax values. While the analysis of our algorithm employs the well-known paradigm of potential-based weighted experts, the learning algorithm itself does not make any use of these weights to devise its predictions. This is in contrast to many learning algorithms that integrate the weights into the prediction process (See [Cesa-Bianchi and Lugosi, 2006, Section 2.1] and bibliographic remarks in Section 2). Can we design a natural algorithm, in the spirit of algorithms such as *weighted majority* [Littlestone and Warmuth, 1994] and Exp4 [Auer, Cesa-Bianchi, Freund, and Schapire, 2002], that achieves the guarantees of Theorem C.7, up to constant factors?

### 4.2 Future research

**The multilabel setting.**    It would be interesting to generalize the results of this work to the multilabel setting considered in [Daniely and Helbertal, 2013], in which the adversary is allowed to choose several correct labels in each round. When the adversary is adaptive, it is not hard to see that this is equivalent to the single-label setting considered in this work with a deterministic learner. What happens when the adversary is oblivious?

**Other types of feedback.**    This work considers the full information and bandit feedback models. However, one can think of other types of feedback. Consider for example the *comparison feedback* model: if the true label is $y$ and the prediction is $z$, the adversary provides the feedback op $\in \{<, =, >\}$ such that $z$ op $y$. An interesting direction for future research is to prove results in the spirit of this work for comparison feedback, or for other natural types of feedback.

## Acknowledgments and Disclosure of Funding

We thank Zachary Chase for many insightful discussions on the problems in the focus of this work.

Shay Moran is a Robert J. Shillman Fellow; he acknowledges support by ISF grant 1225/20, by BSF grant 2018385, by an Azrieli Faculty Fellowship, by Israel PBC-VATAT, by the Technion Center for Machine Learning and Intelligent Systems (MLIS), and by the the European Union (ERC, GENERALIZATION, 101039692). Views and opinions expressed are however those of the author(s) only and do not necessarily reflect those of the European Union or the European Research Council Executive Agency. Neither the European Union nor the granting authority can be held responsible for them.

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

# A  Definitions

Let $\mathcal{X}$ be a *domain*, and let $\mathcal{Y}$ be a countable *label set*. Unless stated otherwise, we assume that $\mathcal{Y} = [k]$ for some natural $k \geq 2$. A pair $(x, y) \in \mathcal{X} \times \mathcal{Y}$ is called an *example*, and an element $x \in \mathcal{X}$ is called an *instance* or an *unlabeled example*. A function $h \colon \mathcal{X} \to \mathcal{Y}$ is called a *hypothesis* or a *concept*. A *hypothesis class*, or *concept class*, is a non-empty set $\mathcal{H} \subset \mathcal{Y}^{\mathcal{X}}$. We note here that it is generally assumed that there exists an instance $x \in \mathcal{X}$ such that for every $y \in \mathcal{Y}$ there exists $h \in \mathcal{H}$ satisfying $h(x) = y$. This assumption is reasonable since when this is not the case, the hypothesis class $\mathcal{H}$ has an essentially isomorphic class $\mathcal{H}'$ with label set strictly smaller than $\mathcal{Y}$ (see [Raman, Raman, Subedi, Mehalel, and Tewari, 2023]).

We will mostly consider the more general notion of *pattern classes*. A *pattern* $p \in (\mathcal{X} \times \mathcal{Y})^{\star}$ is a finite sequence of examples. A pattern class is a non-empty set of patterns $\mathcal{P}$ which is *downwards closed*, meaning that for every $p \in \mathcal{P}$, every sub-pattern $p'$ of $p$ (obtained by removing examples arbitrarily) is in $\mathcal{P}$ as well. Let $p, q$ be sequences (which can be either patterns or some other type of sequences). we denote by $p_I$ the sub-sequence of $p$ consisting only of the indices in $I$. The set $I$ might be given by a clear and well-known notation. For example, if $p = x_1, \ldots, x_T$, then for some $t \leq T$, the notation $p_{\leq t}$ represents the sub-sequence $x_1, \ldots, x_t$. Denote the concatenation of $p, q$ by $p \circ q$. If $p$ is a sub-sequence of $q$, we denote $p \subset q$.

Note that pattern classes generalize concept classes: for every concept class $\mathcal{H}$ we may define the *induced* pattern class

$$\mathcal{P}(\mathcal{H}) = \left\{ p \in (\mathcal{X} \times \mathcal{Y})^{\star} : \min_{h \in \mathcal{H}} \sum_{(x,y) \in p} \mathbb{1}[h(x) \neq y] = 0 \right\}. \tag{7}$$

In words, $\mathcal{P}(\mathcal{H})$ contains all patterns that are consistent with $\mathcal{H}$, and thus from an online-learning perspective there is no essential difference between $\mathcal{H}$ and $\mathcal{P}(\mathcal{H})$. Therefore, throughout this section we consider only pattern classes.

In this work, we consider *multiclass online learning* of pattern classes in the *realizable* setting. Online learning is a repeated game between a learner and an adversary. Each round $t$ in the game proceeds as follows:

(i) The adversary sends an instance $x_t \in \mathcal{X}$ to the learner.

(ii) The learner predicts $\hat{y}_t \in \mathcal{Y}$ (possibly at random).

(iii) The adversary provides (full-information or bandit) feedback to the learner.

*Full-information* feedback means that the learner learns the correct label $y_t$ at the end of every round. *Bandit* feedback means that the learner only receives an indication whether $\hat{y}_t = y_t$ or not. If the learner predicts $\hat{y}_t$ deterministically for every $t$, then the learner is *deterministic*.

In the bandit feedback setting, we model *learners* as functions $\mathsf{Lrn} \colon ((\mathcal{X} \times \{\to, \nrightarrow\} \times \mathcal{Y})^{\star} \times \mathcal{X}) \to \Pi[\mathcal{Y}]$ from the set of pairs of a feedback sequence and an instance, to the set $\Pi[\mathcal{Y}]$ of all probability distributions over $\mathcal{Y}$. For $\pi \in \Pi[\mathcal{Y}]$ we denote the probability of $y$ in $\pi$ by $\pi_y$. In a feedback sequence $(x_1, \mathsf{op}_1, \hat{y}_1), \ldots, (x_T, \mathsf{op}_T, \hat{y}_T)$, the triplet $(x_t, \mathsf{op}_t, \hat{y}_t)$ represents the fact that in round $t$ the instance was $x_t$, and that $\hat{y}_t = y_t$ if $\mathsf{op} = \to$, or $\hat{y}_t \neq y_t$ if $\mathsf{op} = \nrightarrow$. The learner's predictions thus may depend on past feedback as well as on past and current instances. In the full-information feedback model, $\mathsf{op} = \to$ at all times, and so we omit it and use the pair $(x_t, y_t)$ instead of the triplet $(x_t, \mathsf{op}, \hat{y}_t)$.

Given a learner $\mathsf{Lrn}$ and a fixed input sequence of examples $S = (x_1, y_1), \ldots, (x_T, y_T)$, we denote the expected number of mistakes that $\mathsf{Lrn}$ makes when executed on the sequence $S$ by $\mathsf{M}_{\mathrm{full}}(\mathsf{Lrn}; S)$ if it receives full-information feedback, and by $\mathsf{M}_{\mathrm{bandit}}(\mathsf{Lrn}; S)$ if it receives bandit feedback. We define the optimal mistake bound of $\mathcal{P}$ when the adversary is *oblivious* to be

$$\mathsf{opt}_{\mathrm{full}}^{\mathrm{rand}}(\mathcal{P}) = \inf_{\mathsf{Lrn}} \sup_{S \in \mathcal{P}} \mathsf{M}_{\mathrm{full}}(\mathsf{Lrn}; S), \quad \mathsf{opt}_{\mathrm{bandit}}^{\mathrm{obl}}(\mathcal{P}) = \inf_{\mathsf{Lrn}} \sup_{S \in \mathcal{P}} \mathsf{M}_{\mathrm{bandit}}(\mathsf{Lrn}; S), \tag{8}$$

in the full information and bandit feedback models, respectively.

**Remark A.1** (Strong vs. Weak Realizability)**.** *The restriction $S \in \mathcal{P}$ that the supremum is taken over is called strong realizability, which requires that the input sequence is taken from the class $\mathcal{P}$. On*

*the other hand, unless stated otherwise, in this work we consider the weak realizability assumption. Under this assumption, it is only required that for any learner, the feedback provided by the adversary is consistent with some pattern from $\mathcal{P}$ with probability 1. When the adversary is oblivious and the input sequence $S$ is chosen beforehand, strong and weak realizability coincide. Indeed, note that for any learner who never predicts a label with probability 0, strong realizability must hold in order to satisfy weak realizability.*

The adversary in the above definition is indeed oblivious to the learner's actions, in the sense that it must choose the *target pattern* $S \in \mathcal{P}$ before the beginning of the game. In contrast, in the setting of the bandit feedback model we will also consider a stronger *adaptive* adversary which is allowed to choose the target pattern on the fly, as long as it satisfies the weak realizability assumption, meaning that the feedback provided to the learner is consistent with some $p \in \mathcal{P}$ with probability 1. In addition, it is allowed to choose the correct label at random.[6]

When the adversary is adaptive, the online learning game defined above operates as follows in every round $t$:

  (i) The adversary sends an instance $x_t \in \mathcal{X}$ to the learner.

 (ii) The learner chooses a probability distribution $\pi^{(t)}$ over $\mathcal{Y}$ and reveals it to the adversary.

 (iii) The adversary chooses a probability distribution $\tau^{(t)}$ over $\mathcal{Y}$.

 (iv) The learner draws a prediction $\hat{y}_t \in \mathcal{Y}$ from $\pi^{(t)}$ and reveals it to the adversary.

 (v) The adversary draws the correct label $y_t$ from $\tau^{(t)}$ and tells the learner if its prediction was correct or not.

We formalize an adaptive adversary as a pair of functions

$$\mathsf{Adv}_x \colon (\mathcal{X} \times \{\rightarrow, \nrightarrow\} \times \mathcal{Y})^\star \rightarrow \mathcal{X}, \quad \mathsf{Adv}_y \colon (\mathcal{X} \times \{\rightarrow, \nrightarrow\} \times \mathcal{Y})^\star \times \mathcal{X} \times \Pi[\mathcal{Y}] \rightarrow \Pi[\mathcal{Y}].$$

In every round $t$, $\mathsf{Adv}_x$ is used in the first step to choose the instance $x_t$, and $\mathsf{Adv}_y$ is used in the third step to determine the distribution from which the correct label $y_t \in \mathcal{Y}$ is drawn. In more detail, let $F \in (\mathcal{X} \times \{\rightarrow, \nrightarrow\} \times \mathcal{Y})^\star$ be the feedback sequence given to the learner before round $t$. The adversary sends the instance $x_t = \mathsf{Adv}_x(F)$ to the learner in the first step. In the third step, the adversary computes the distribution $\tau^{(t)} = \mathsf{Adv}_y(F, x_t, \pi^{(t)})$.

A feedback sequence $F \in (\mathcal{X} \times \{\rightarrow, \nrightarrow\} \times \mathcal{Y})^\star$ is *realizable* by $\mathcal{P}$ if there exists a pattern in $\mathcal{P}$ that is consistent with $F$. An adaptive adversary is *consistent* with $\mathcal{P}$ if for every feedback sequence $F \in (\mathcal{X} \times \{\rightarrow, \nrightarrow\} \times \mathcal{Y})^\star$ which is realizable by $\mathcal{P}$ and for every $\pi \in \Pi[\mathcal{Y}]$, the following holds. Let $x = \mathsf{Adv}_x(F)$. If $\mathsf{Adv}_y(F, x, \pi) = \tau$, then the feedback sequence $F \circ (x, \rightarrow, y)$ is realizable for every $y$ in the support of $\tau$.

Given a learner $\mathsf{Lrn}$ and an adaptive adversary $\mathsf{Adv}$ which provides bandit feedback, we denote the expected number of mistakes that $\mathsf{Lrn}$ makes against $\mathsf{Adv}$ by $\mathsf{M}_{\mathrm{bandit}}(\mathsf{Lrn}; \mathsf{Adv})$. We define the optimal mistake bound of $\mathcal{P}$ against an adaptive adversary which provides bandit feedback by

$$\mathsf{opt}^{\mathrm{adap}}_{\mathrm{bandit}}(\mathcal{P}) = \inf_{\mathsf{Lrn}} \sup_{\mathsf{Adv}} \mathsf{M}(\mathsf{Lrn}; \mathsf{Adv}), \tag{9}$$

where the supremum is taken over all adaptive adversaries which are consistent with $\mathcal{P}$.

We denote by $\mathsf{opt}^{\mathrm{det}}_{\mathrm{full}}(\mathcal{P})$ and $\mathsf{opt}^{\mathrm{det}}_{\mathrm{bandit}}(\mathcal{P})$ the optimal deterministic mistake bounds[7] of $\mathcal{P}$ with full-information and bandit feedback, respectively. That is, in these settings we require the additional restriction that $\mathsf{Lrn}$ must be deterministic. We may sometimes refer to mistake bounds as the learner's *loss* throughout the entire game. Unless stated otherwise, the notations $O, \Omega, \Theta$ hide universal constants which do not depend on any parameter of the problem.

## B The primal and dual games

In this section we describe a game-theoretic formulation of $\mathsf{opt}^{\mathrm{adap}}_{\mathrm{bandit}}$, and apply the minimax theorem to obtain a dual formulation which will be useful in Section C. We term the game defined in Section A

---

[6]It is explained in the sequel that being randomized does not really help the adversary

[7]One can verify that in the deterministic case, oblivious and adaptive adversaries are equivalent.

when played against an adaptive adversary as the *primal game*. In the *dual game*, we basically just switch the order of the second and third steps of the primal game (together with some other required but not meaningful minor changes). We can think of the primal and dual games as the same game, only that after the adversary chooses $x_t$, in the primal game the learner chooses a distribution (over the labels) first, and in the dual game the adversary chooses a distribution first. Formally, the dual game operates as follows in each round $t$:

(i) The adversary picks an instance $x_t \in \mathcal{X}$ and a probability distribution $\tau^{(t)}$ over $\mathcal{Y}$. It reveals both to the learner.

(ii) The learner chooses a probability distribution $\pi^{(t)}$ over $\mathcal{Y}$, draws a prediction $\hat{y}_t \in \mathcal{Y}$ from $\pi^{(t)}$, and reveals it to the adversary.

(iii) The adversary draws the correct label $y_t$ from $\tau^{(t)}$ and tells the learner if its prediction was correct or not.

In this section, we show that the primal and dual games are equivalent in terms of their optimal mistake bounds. The motivation behind this is that the target function of the optimization problem describing the optimal mistake bound in the dual game will turn out to be simpler. This will enable the analysis in Section C, and might be of further interest. The equivalence will mainly follow from von Neumann's minimax theorem [von Neumann, 1928]. Using minimax duality to analyze minimax target functions of online learning problems has proved useful in previous works. Notable examples are [Abernethy, Agarwal, Bartlett, and Rakhlin, 2009, Rakhlin, Sridharan, and Tewari, 2010, Rakhlin, Shamir, and Sridharan, 2012].

We now get into the details. In the dual game, $\pi^{(t)}$ depends on the distribution $\tau^{(t)}$ chosen by the adversary, and the distribution $\tau^{(t)}$ does not depend on the distribution $\pi^{(t)}$. Therefore we have to change the formal definitions of a learner and an adversary in the dual game. The adversary is defined as a single function $\mathsf{Adv}\colon (\mathcal{X} \times \{\rightarrow, \nrightarrow\} \times \mathcal{Y})^\star \to (\mathcal{X} \times \Pi[\mathcal{Y}])$. The learner is defined as a function $\mathsf{Lrn}\colon (\mathcal{X} \times \{\rightarrow, \nrightarrow\} \times \mathcal{Y})^\star \times \mathcal{X} \times \Pi[\mathcal{Y}] \to \Pi[\mathcal{Y}]$. The optimal mistake bounds in the primal and dual games are defined in a dual manner:

$$\mathsf{opt}^{\mathrm{adap}}_{\mathrm{bandit}}(\mathcal{P}) = \inf_{\mathsf{Lrn}} \sup_{\mathsf{Adv}} \mathsf{M}(\mathsf{Lrn};\mathsf{Adv}), \quad \mathsf{opt\_d}^{\mathrm{adap}}_{\mathrm{bandit}}(\mathcal{P}) = \sup_{\mathsf{Adv}} \inf_{\mathsf{Lrn}} \mathsf{M}(\mathsf{Lrn};\mathsf{Adv}). \quad (10)$$

First, we provide explicit definitions for $\mathsf{opt}^{\mathrm{adap}}_{\mathrm{bandit}}(\mathcal{P})$ and $\mathsf{opt\_d}^{\mathrm{adap}}_{\mathrm{bandit}}(\mathcal{P})$ in terms of simpler classes, based on the instance and feedback provided by the adversary in every round. For every instance $x$ define the class:

$$\mathcal{P}_x = \{p \in \mathcal{P} : (x, y) \circ p \in \mathcal{P} \text{ for some } y \in \mathcal{Y}\}.$$

Additionally, define the following classes for every pair of an instance and a label $(x, y)$:

$$\mathcal{P}_{x \rightarrow y} = \{p \in \mathcal{P} : (x, y) \circ p \in \mathcal{P}\}, \quad \mathcal{P}_{x \nrightarrow y} = \mathcal{P}_x \backslash \mathcal{P}_{x \rightarrow y}.$$

To simplify presentation, we assume that in the primal game the adversary is deterministic, and in the dual game the learner is deterministic. This is reasonable since after the first player has made its choice, the optimal choice of the second player is deterministic. Let us now define $\mathsf{opt}^{\mathrm{adap}}_{\mathrm{bandit}}(\mathcal{P})$ and $\mathsf{opt\_d}^{\mathrm{adap}}_{\mathrm{bandit}}(\mathcal{P})$ in terms of classes of the form $\mathcal{P}_{x \rightarrow y}$ and $\mathcal{P}_{x \nrightarrow y}$:

$$\mathsf{opt}^{\mathrm{adap}}_{\mathrm{bandit}}(\mathcal{P}) = \sup_x \inf_\pi \sup_{y \in \mathcal{Y}} \left[ \pi_y \cdot \mathsf{opt}^{\mathrm{adap}}_{\mathrm{bandit}}(\mathcal{P}_{x \rightarrow y}) + \sum_{y' \neq y} \pi_{y'} \left( 1 + \mathsf{opt}^{\mathrm{adap}}_{\mathrm{bandit}}(\mathcal{P}_{x \nrightarrow y'}) \right) \right], \quad (11)$$

$$\mathsf{opt\_d}^{\mathrm{adap}}_{\mathrm{bandit}}(\mathcal{P}) = \sup_{x,\tau} \inf_{y \in \mathcal{Y}} \left[ \tau_y \cdot \mathsf{opt\_d}^{\mathrm{adap}}_{\mathrm{bandit}}(\mathcal{P}_{x \rightarrow y}) + (1 - \tau_y) \left( 1 + \mathsf{opt\_d}^{\mathrm{adap}}_{\mathrm{bandit}}(\mathcal{P}_{x \nrightarrow y}) \right) \right]. \quad (12)$$

For these equations to be well-defined, it is technically convenient to define $\mathsf{opt}^{\mathrm{adap}}_{\mathrm{bandit}}(\emptyset) = \mathsf{opt\_d}^{\mathrm{adap}}_{\mathrm{bandit}}(\emptyset) = -\infty$. Now, by the law of total expectation, for every pattern class $\mathcal{P}$ the above equations indeed define the optimal losses in the primal and dual games. We can now prove the equivalence.

**Lemma B.1.** *For every pattern class $\mathcal{P}$:*

$$\mathsf{opt}^{\mathrm{adap}}_{\mathrm{bandit}}(\mathcal{P}) = \mathsf{opt\_d}^{\mathrm{adap}}_{\mathrm{bandit}}(\mathcal{P}).$$

*Proof.* For a horizon (number of rounds) $T$, let $\mathsf{opt}^{\mathrm{adap}}_{\mathrm{bandit}}(\mathcal{P}, T), \mathsf{opt\_d}^{\mathrm{adap}}_{\mathrm{bandit}}(\mathcal{P}, T)$ be the analogue notations to $\mathsf{opt}^{\mathrm{adap}}_{\mathrm{bandit}}(\mathcal{P}), \mathsf{opt\_d}^{\mathrm{adap}}_{\mathrm{bandit}}(\mathcal{P})$, with the additional restriction that the game is played for $T$ rounds. We adapt (11) and (12) to those definitions in the obvious way. Note that

$$\mathsf{opt}^{\mathrm{adap}}_{\mathrm{bandit}}(\mathcal{P}) = \sup_{T \in \mathbb{N}, x \in \mathcal{X}} \mathsf{opt}^{\mathrm{adap}}_{\mathrm{bandit}}(\mathcal{P}_x, T),$$

and similarly for $\mathsf{opt\_d}^{\mathrm{adap}}_{\mathrm{bandit}}(\mathcal{P})$. Therefore, to prove the lemma, it suffices to show that $\mathsf{opt}^{\mathrm{adap}}_{\mathrm{bandit}}(\mathcal{P}_x, T) = \mathsf{opt\_d}^{\mathrm{adap}}_{\mathrm{bandit}}(\mathcal{P}_x, T)$ for all $x, T$ such that $\mathcal{P}_x \neq \emptyset$ (when $\mathcal{P}_x = \emptyset$ both are $-\infty$ for all $T$). Let $x \in \mathcal{X}$ be such that $\mathcal{P}_x \neq \emptyset$. For $T = 0$ we have $\mathsf{opt}^{\mathrm{adap}}_{\mathrm{bandit}}(\mathcal{P}_x, 0) = 0 = \mathsf{opt\_d}^{\mathrm{adap}}_{\mathrm{bandit}}(\mathcal{P}_x, 0)$.

For the induction step, note that the expression for $\mathsf{opt\_d}^{\mathrm{adap}}_{\mathrm{bandit}}(\mathcal{P}_x, T)$ suggested by (12) describes a zero-sum game in which the adversary goes first, and both the learner and adversary choose distributions over $\mathcal{Y}$ to draw a label from: the learner draws a prediction and the adversary draws the correct label. The target function is the optimal loss to be suffered by the learner in the sequel, after the distributions are fixed. The adversary's goal is to maximize it and the learner's goal is to minimize it. Since this is a zero-sum game, by the minimax theorem [von Neumann, 1928] we can let the learner go first instead, without changing the value of $\mathsf{opt\_d}^{\mathrm{adap}}_{\mathrm{bandit}}(\mathcal{P}_x, T)$. Therefore we can write $\mathsf{opt\_d}^{\mathrm{adap}}_{\mathrm{bandit}}(\mathcal{P}_x, T)$ as:

$$\inf_{\pi} \sup_{y \in \mathcal{Y}} \left[ \pi_y \cdot \mathsf{opt\_d}^{\mathrm{adap}}_{\mathrm{bandit}}(\mathcal{P}_{x \to y}, T - 1) + \sum_{y' \neq y} \pi_{y'}(1 + \mathsf{opt\_d}^{\mathrm{adap}}_{\mathrm{bandit}}(\mathcal{P}_{x \not\to y'}, T - 1)) \right].$$

By the induction hypothesis, we can change every $\mathsf{opt\_d}^{\mathrm{adap}}_{\mathrm{bandit}}$ to $\mathsf{opt}^{\mathrm{adap}}_{\mathrm{bandit}}$, which gives:

$$\inf_{\pi} \sup_{y \in \mathcal{Y}} \left[ \pi_y \cdot \mathsf{opt}^{\mathrm{adap}}_{\mathrm{bandit}}(\mathcal{P}_{x \to y}, T - 1) + \sum_{y' \neq y} \pi_{y'}(1 + \mathsf{opt}^{\mathrm{adap}}_{\mathrm{bandit}}(\mathcal{P}_{x \not\to y'}, T - 1)) \right]. \tag{13}$$

By applying (13), observe that $\mathsf{opt\_d}^{\mathrm{adap}}_{\mathrm{bandit}}(\mathcal{P}_x, T) = \mathsf{opt}^{\mathrm{adap}}_{\mathrm{bandit}}(\mathcal{P}_x, T)$, as required. $\square$

## B.1 An optimal learner for the primal game

The optimal randomized algorithm achieving $\mathsf{opt}^{\mathrm{adap}}_{\mathrm{bandit}}(\mathcal{P})$ suggests itself from (11). For completeness, we explicitly write it in Figure 1. The algorithm may be implemented using straightforward dynamic programming for finite hypothesis classes, even in the $r$-realizable case where the best hypothesis is inconsistent with the feedback in at most $r$ many indices.

**Proposition B.2.** *The algorithm* BanditRandSOA *is well-defined and optimal.*

*Proof.* The choice of $\pi^{(t)}$ is well defined because (14) is a feasible linear optimization problem, and as such it has a solution. The algorithm is optimal due to (11). $\square$

# C Prediction with expert advice

In this section we are interested in optimal mistake bounds for the bandit feedback setup of *prediction with expert advice* in the $r$-realizable setting. In this problem, $n$ experts are making predictions in each round, and it is promised that the best expert is inconsistent with the adversary's feedback in at most $r$ many rounds. An adversary following this limitation is called $r$-*consistent*. Nothing else is assumed about how the experts decide on their predictions. Furthermore, in Section G we explain how to remove the assumption that $r$ is given to the learner with only a constant factor degradation in the mistake bound. Based on the experts' predictions, the learner should make its own prediction for each round, while minimizing the expected number of mistakes it makes in the entire game. In



**BanditRandSOA**

**Input:** A pattern class $\mathcal{P}$.

**For** $t = 1, 2, \ldots$

    1. Receive instance $x_t \in \mathcal{X}$.

    2. Construct a distribution $\pi^{(t)}$ such that

$$\pi^{(t)} \in \arg\min_{\pi \in \Pi[\mathcal{Y}]} \max_{y \in \mathcal{Y}} \left[ \pi_y \mathsf{opt}_{\mathrm{bandit}}^{\mathrm{adap}}(\mathcal{P}_{x \to y}) + \sum_{y' \neq y} \pi_{y'}(1 + \mathsf{opt}_{\mathrm{bandit}}^{\mathrm{adap}}(\mathcal{P}_{x \not\to y'})) \right].$$
$$(14)$$

    3. Draw the prediction $\hat{y}_t$ from $\pi^{(t)}$.

    4. If the feedback is positive, update $\mathcal{P} := \mathcal{P}_{x \to \hat{y}_t}$, and otherwise update $\mathcal{P} := \mathcal{P}_{x \not\to \hat{y}_t}$.



Figure 1: BanditRandSOA is an optimal randomized learner for online learning with bandit feedback of pattern classes, where the adversary is allowed to be adaptive. It is inspired by the RandSOA algorithm of Filmus, Hanneke, Mehalel, and Moran [2023], which is a randomized variant of Littlestnoe's [Littlestone, 1988] well-known SOA algorithm.

the binary or full-information feedback model, this problem is well-studied, and tight bounds were proved for both deterministic and randomized learners [Vovk, 1990, Littlestone and Warmuth, 1994, Cesa-Bianchi, Freund, Helmbold, and Warmuth, 1996, Cesa-Bianchi, Freund, Haussler, Helmbold, Schapire, and Warmuth, 1997, Abernethy, Langford, and Warmuth, 2006, Brânzei and Peres, 2019, Filmus, Hanneke, Mehalel, and Moran, 2023].

In this work, we are mostly interested in the randomized (learner) and adaptive (adversary) setup with bandit feedback, and the bound we prove for it plays a central role in the proof of Theorem 1.1. For the sake of better completeness and exhaustiveness, we consider the other deterministic (learner) and oblivious (adversary) setups as well. Our results for all three setups are summarized in Table 2.

We adapt the notation for learning pattern classes. Suppose that there are $k \geq 2$ many labels and $n \geq k$ many experts, where the best expert is inconsistent with the feedback in at most $r \geq 0$ many rounds. We assume that $n \geq k$, since if $n < k$ then having $k$ labels is equivalent to having $n$ labels. When the learner is deterministic, the optimal mistake bound is denoted by $\mathsf{opt}_{\mathrm{bandit}}^{\mathrm{det}}(n, k, r)$. When the learner is randomized, the optimal mistake bound is denoted by $\mathsf{opt}_{\mathrm{bandit}}^{\mathrm{obl}}(n, k, r)$ or $\mathsf{opt}_{\mathrm{bandit}}^{\mathrm{adap}}(n, k, r)$, depending on whether the adversary is oblivious or adaptive, respectively. For the realizable case $r = 0$, we omit $r$ from the notation.

## C.1 Warm-up: deterministic learners

In this section, we analyze $\mathsf{opt}_{\mathrm{bandit}}^{\mathrm{det}}(n, k, r)$ and prove the following theorem.

**Theorem C.1.** *Let $n \geq k \geq 2, r \geq 0$. Then:*

$$\mathsf{opt}_{\mathrm{bandit}}^{\mathrm{det}}(n, k, r) = \Theta(k(\log(n/k) + r + 1)).$$

We first prove the upper bound and then the lower bound.

**Lemma C.2.** *Let $n \geq k \geq 2, r \geq 0$. Then for every $\alpha > 1$:*

$$\mathsf{opt}_{\mathrm{bandit}}^{\mathrm{det}}(n, k, r) \leq \frac{\alpha}{\alpha - 1} k \ln(n \cdot \alpha^r / k) + k - 1.$$

*Proof.* We describe a deterministic learner which makes at most the stated number of mistakes. We use a simple generalization of *Weighted Majority* [Littlestone and Warmuth, 1994]. A similar idea appears also in [Long, 2020].

Every expert is given an initial weight of $\alpha^r$. Let $W_t$ be the total weight of experts in round $t$. In every round, predict the label that enjoys a weighted plurality. If we make a mistake, then the weight of every expert which voted for the predicted label is divided by $\alpha$. If an expert reaches weight less than $1$, we reduce its weight to $0$, since it must mean that it made more than $r$ mistakes. We split the analysis into two epochs. The first epoch is as long as $W_t > k$, and the second is afterwards.

We begin by analyzing the first epoch. By assumption, $W_t > k$ at all times. On the other hand, whenever the learner makes a mistake, at least $1/k$ of the weight is being divided by $\alpha$. Therefore, if a mistake occurs in round $t$ then

$$W_{t+1} \leq (1 - 1/k)W_t + W_t/(\alpha k) = \left(1 - \frac{\alpha - 1}{\alpha k}\right)W_t.$$

Now, if $m_1$ is the total number of mistakes the learner makes as long as $W_t > k$, then $m_1$ must satisfy

$$k < n \cdot \alpha^r \left(1 - \frac{\alpha - 1}{\alpha k}\right)^{m_1} \leq n \cdot \alpha^r e^{-m_1 \frac{\alpha - 1}{\alpha k}}.$$

After some manipulations, we derive the bound

$$m_1 < \frac{\alpha k}{\alpha - 1} \ln(n \cdot \alpha^r / k).$$

We now analyze $m_2$, which is the number of mistakes in the second epoch. If $W_t \leq k$ then there are at most $k$ experts which have not yet made more than $r$ mistakes. At least one of them will never err again, and therefore in the worst case, the learner will make $m_2 \leq k - 1$ more mistakes before eliminating all experts apart from the target expert. Summing $m_1 + m_2$ gives the stated bound. $\square$

We may now deduce the upper bound.

**Corollary C.3.** *Let $n \geq k \geq 2, r \geq 0$. Then:*

$$\mathrm{opt}_{\mathrm{bandit}}^{\mathrm{det}}(n, k, r) \leq \frac{e}{e - 1} k(\ln(n/k) + r) + k - 1.$$

*Proof.* Apply Lemma C.2 with $\alpha = e$. $\square$

We can also deduce an improved upper bound for the realizable ($r = 0$) case.

**Corollary C.4.** *Let $n \geq k \geq 2$. Then:*

$$\mathrm{opt}_{\mathrm{bandit}}^{\mathrm{det}}(n, k) \leq k \ln(n/k) + k - 1.$$

*Proof.* We can easily see that $\mathrm{opt}_{\mathrm{bandit}}^{\mathrm{det}}(n, k) \leq (1+\epsilon)k \ln(n/k) + k - 1$ for every $\epsilon > 0$ by applying Lemma C.2 with $\alpha \to \infty$. There is also an explicit optimal learner for which this holds with $\epsilon = 0$, obtained by slightly changing the proof of Lemma C.2. $\square$

We now turn to the lower bounds. We begin with the realizable case.

**Lemma C.5.** *Let $n \geq k \geq 3$. Then:*

$$\mathrm{opt}_{\mathrm{bandit}}^{\mathrm{det}}(n, k) \geq (k/2 - 1) \ln(n/k).$$

*Proof.* The adversary uses the following simple strategy. Let $n_t$ be the number of alive experts (that is, experts which are consistent with the feedback given so far) at the beginning of round $t$. As long as $n_t > k$, in every round $t$ the adversary splits the alive experts as evenly as possible among the $k$ labels, and gives negative feedback to the learner. Note that the largest set in the partition consists of $\lceil n_t/k \rceil \leq n_t/k + 1 < 2n_t/k$ experts, since $n_t > k$. Therefore, in every round at least a $(1 - 2/k)$-fraction of the experts make it to the next round. The adversary can thus force at least $m$ many mistakes on the learner whenever $m$ satisfies

$$(1 - 2/k)^{m-1}n > k,$$

which holds if

$$e^{-\frac{2}{k-2}(m-1)}n > k,$$

by using the inequality $1 - x > e^{-\frac{x}{x-1}}$ that holds for all $x < 1$. After some manipulations, the inequality reads

$$m \leq (k/2 - 1)\ln(n/k) + 1,$$

which implies the stated bound. $\qquad\square$

To devise a lower bound for the $r$-realizable setting, we prove the following lemma. It was independently proved also by Geneson and Tang [2024].

**Lemma C.6.** *For every $n \geq k \geq 2, r \geq 0$ we have:*

$$\mathsf{opt}_{\mathrm{bandit}}^{\mathrm{det}}(n, k, r) \geq k(r + 1) - 1.$$

*Proof.* Suppose that there are $n = k$ experts. In every round for $T = k(r + 1) - 1$ rounds, the adversary lets expert $i$ predict the label $i$, and always provides negative feedback to the learner. Let $i^\star$ be the label that the learner predicts the least number of times, and let $i' \neq i^\star$. Since $T = k(r+1)-1$, it must hold that $i^\star$ is predicted by the learner at most $r$ times. In every round in which the learner predicts $i^\star$, set the true label to $i'$. In all other rounds set the true label to $i^\star$. By definition of the strategy, the best expert makes at most $r$ mistakes. $\qquad\square$

We may now prove Theorem C.1.

*Proof of Theorem C.1.* The stated bounds are known when $k = 2$ by e.g. [Vovk, 1990, Littlestone and Warmuth, 1994]. For $k \geq 3$, the upper bound follows from Corollary C.3, and the lower bound follows from Lemmas C.5 and C.6. $\qquad\square$

## C.2 Randomized (Learner) and Adaptive (Adversary)

In this section we provide a tight upper bound on $\mathsf{opt}_{\mathrm{bandit}}^{\mathrm{adap}}(n, k, r)$ by proving the following theorem.

**Theorem C.7.** *Let $n \geq k \geq 2, r \geq 0$. Then:*

$$\mathsf{opt}_{\mathrm{bandit}}^{\mathrm{adap}}(n, k, r) \leq k \log_k n + 2kr.$$

We will prove the theorem by analyzing the mistake bound of the equivalent dual game, as described in Section B. The primal and dual games are described in Section B in terms of pattern classes, so we first show that the problem of prediction with expert advice in the $r$-realizable setting is in fact equivalent to learning an appropriate specific pattern class.

For every hypothesis class $\mathcal{H}$ and *budget function* $B \colon \mathcal{H} \to \mathbb{R}$, we define the pattern class

$$\mathcal{P}(\mathcal{H}, B) = \left\{ p \in (\mathcal{X} \times \mathcal{Y})^\star : \left[ \exists h \in \mathcal{H} : \sum_{(x,y) \in p} \mathbb{1}[h(x) \neq y] \leq B(h) \right] \right\}. \tag{15}$$

In words, $\mathcal{P}(\mathcal{H}, B)$ consists precisely of those patterns which are consistent with some $h \in \mathcal{H}$ everywhere except for at most $B(h)$ many places. We call $B(h)$ the *budget* of $h$. Therefore, if $B_r$ is the budget function that gives budget $r$ to all hypotheses, then learning the pattern class $\mathcal{P}(\mathcal{H}, B_r)$ under the assumption of realizability is equivalent to learning the hypothesis class $\mathcal{H}$ under the assumption of $r$-realizability.

Observe that for a fixed number of labels $k$, the class of $n$ experts is simply the hardest concept class of size $n$, since every expert can predict any label in every round. Denote this class by $\mathcal{U}_{n,k}$, so that *prediction with expert advice* with $n$ experts and $k$ labels in the $r$-realizable setting is equivalent to learning the pattern class $\mathcal{P}(\mathcal{U}_{n,k}, B_r)$ under the assumption of realizability. Therefore, we have

$$\mathsf{opt}_{\mathrm{bandit}}^{\mathrm{adap}}(n, k, r) = \mathsf{opt}_{\mathrm{bandit}}^{\mathrm{adap}}(\mathcal{P}(\mathcal{U}_{n,k}, B_r)) = \mathsf{opt\_d}_{\mathrm{bandit}}^{\mathrm{adap}}(\mathcal{P}(\mathcal{U}_{n,k}, B_r))$$

due to Lemma B.1.

For completeness, we formally define $\mathcal{U}_{n,k}$. This is the class of projection functions over the set of $k$-ary vectors of length $n$, $\mathcal{X} = [k]^n$. That is, if the $n$ functions are $\{h_1, \ldots, h_n\}$ then for every $x \in \mathcal{X}$ we have $h_i(x) = x_i$, where $x_i$ is the value of the $i$'th entry of $x$.

We now define some notation used in the proof. As observed in [Abernethy, Langford, and Warmuth, 2006, Brânzei and Peres, 2019], since the experts are identical, in every round of the game we only care about how many of the experts are still *alive* (which means they have not been inconsistent with the feedback for more than $r$ many rounds), and among those which are alive, how many have budget $i$, for each $0 \leq i \leq r$. Following [Abernethy, Langford, and Warmuth, 2006, Brânzei and Peres, 2019], we represent this information by an $(r+1)$-ary vector $\vec{m} = (m_0, \ldots, m_r)$, in which the number $m_i$ indicates the number of living experts with budget $i$. We call this vector the *state* of the game.

For any state $\vec{m}$, let $V(\vec{m})$ be the optimal loss in the dual (or primal) game for the pattern class representing the state $\vec{m}$, as defined in (15). That is, if $\mathcal{P}(\mathcal{U}_{n,k}, B_{\vec{m}})$ is the pattern class representing the state $\vec{m}$, then $V(\vec{m}) = \mathsf{opt\_d}_{\mathrm{bandit}}^{\mathrm{adap}}(\mathcal{P}(\mathcal{U}_{n,k}, B_{\vec{m}}))$. We call $V(\vec{m})$ the *value* of the state $\vec{m}$. The goal is thus to upper bound the value of the initial state $V(0, 0, \ldots, n)$.

Towards this end, we use the following potential-based technique. An expert with budget $i$ will have a potential of $k^{2i}$. This will allow us to bound $V(\vec{m})$ in terms of the total potential of the experts, given by $W(\vec{m}) = \sum_{i=0}^{r} m_i \cdot k^{2i}$.

Let $b(k) = \frac{k^k}{(k-1)^{k-1}}$. We will prove the the following.

**Lemma C.8.** *Let $k \geq 2$ be the number of labels. For any state $\vec{m}$ it holds that*
$$V(\vec{m}) \leq k \cdot \log_{b(k)} W(\vec{m}).$$

In order to prove Lemma C.8, we will need the following technical claim, proved at the end of this section.

**Lemma C.9.** *Let $k \geq 2$, and let*
$$f_k(\beta) := \log_{b(k)}(((1-\beta)/k^2 + \beta)(\beta/k^2 + (1-\beta))^{k-1}) - 1/k.$$
*Then $f_k(\beta) \leq -1$ for all $\beta \in [0,1]$.*

We can now prove Lemma C.8.

*Proof of Lemma C.8.* The proof is by induction on the state, ordered in the obvious way: the state $(1, 0, \ldots, 0)$ is the lowest, and the state $(0, \ldots, 0, n)$ is the highest.

In order to be able to always apply the induction hypothesis, we must assume that the state always decreases between rounds. Indeed, the only case where the state does not decrease between rounds is when all experts predict the same label $y$. In this case, an optimal adversary must choose $y$ as the correct label with probability 1, and thus an optimal learner must predict $y$. Nothing is changed in such rounds, and therefore they may be removed, ensuring that the induction hypothesis can always be applied.

In the base case, $\vec{m} = (1, 0, \ldots, 0)$. Therefore
$$V(\vec{m}) = 0 = k \log_{b(k)} 1 = k \log_{b(k)} W(\vec{m}).$$

For the induction step, consider the first round of the game, in which the adversary draws the correct label from a fixed distribution $\tau$. Let $V_y(\vec{m})$ be the optimal number of mistakes made by the learner if it predicts $y$ when the state of the game is $\vec{m}$. Let $\vec{m}_{\to y}$ be the state of the game in the next round, if in the current round the correct label is $y$. Define $\vec{m}_{\not\to y}$ in a similar way. Using (12), we may now write
$$V_y(\vec{m}) = \tau_y V(\vec{m}_{\to y}) + (1 - \tau_y)(1 + V(\vec{m}_{\not\to y})).$$
Since the learner is allowed to predict any $y \in \mathcal{Y}$ (and specifically the $y$ which minimizes $V_y(\vec{m})$), it suffices to bound $V_y(\vec{m})$ for some $y$ of our choice. We now consider two cases.

**Case 1.** Suppose that there exists $y \in \mathcal{Y}$ which satisfies
$$V(\vec{m}_{\to y}) \geq 1 + V(\vec{m}_{\not\to y}).$$

Then:
$$V_y(\vec{m}) \leq V(\vec{m}_{\to y}) \leq k \log_{b(k)} W(\vec{m}_{\to y}) \leq k \log_{b(k)} W(\vec{m})$$
by the induction hypothesis, concluding the first case.

**Case 2.** Suppose that for every $y \in \mathcal{Y}$,
$$V(\vec{m}_{\to y}) < 1 + V(\vec{m}_{\not\to y}).$$
Let $y \in \mathcal{Y}$ be such that $\tau_y \geq 1/k$. Then by assumption,
$$V_y(\vec{m}) \leq (1/k)V(\vec{m}_{\to y}) + (1 - 1/k)(1 + V(\vec{m}_{\not\to y})).$$
The induction hypothesis now implies that
$$V_y(\vec{m}) \leq 1 - 1/k + \log_{b(k)} W(\vec{m}_{\to y}) + (k-1)\log_{b(k)} W(\vec{m}_{\not\to y}). \tag{16}$$

Let $\beta \in [0, 1]$ be the fraction of experts predicting $y$, weighted according to their potential. If the true label is $y$ then a $\beta$-fraction of the potential $W(\vec{m})$ is untouched, and a $(1 - \beta)$-fraction is multiplied by at most $1/k^2$ (experts with budget 1 have their potential multiplied by 0), and so
$$W(\vec{m}_{\to y}) \leq ((1 - \beta)/k^2 + \beta)W(\vec{m}).$$
Similarly,
$$W(\vec{m}_{\not\to y}) \leq (\beta/k^2 + (1 - \beta))W(\vec{m}).$$
Plugging this into (16), we have
$$V_y(\vec{m}) \leq 1 - 1/k + \log_{b(k)}((1 - \beta)/k^2 + \beta)W(\vec{m}) + (k-1)\log_{b(k)}(\beta/k^2 + (1 - \beta))W(\vec{m}),$$
which after a bit of manipulation reads as
$$V_y(\vec{m}) \leq k \log_{b(k)} W(\vec{m}) + 1 + \log_{b(k)}(((1 - \beta)/k^2 + \beta)(\beta/k^2 + (1 - \beta))^{k-1}) - 1/k.$$
To finish the proof, it remains to show that
$$f(\beta) := \log_{b(k)}(((1 - \beta)/k^2 + \beta)(\beta/k^2 + (1 - \beta))^{k-1}) - 1/k$$
is bounded from above by $-1$ for all $\beta \in [0, 1]$, which is the statement of Lemma C.9. $\square$

We can now prove Theorem C.7.

*Proof of Theorem C.7.* Clearly $b(k) \geq k$. Applying Lemma C.8 on the initial state $\vec{m} = (0, \ldots, 0, n)$, we get
$$\mathsf{opt}_{\mathrm{bandit}}^{\mathrm{adap}}(n, k, r) = \mathsf{opt\_d}_{\mathrm{bandit}}^{\mathrm{adap}}(\mathcal{P}(\mathcal{U}_{n,k}, B_r)) = V(0, \ldots, 0, n) \leq k \log_{b(k)}(n \cdot k^{2r}) \leq k \log_k n + 2kr,$$
as desired. $\square$

### C.2.1 Proof of technical lemma

In this section we prove Lemma C.9. Let us first restate it in a more convenient way. Define
$$g_k(\beta) = \left(\frac{1-\beta}{k^2} + \beta\right)\left(\frac{\beta}{k^2} + 1 - \beta\right)^{k-1}.$$
The lemma states that for all $k \geq 2$ and $\beta \in [0, 1]$, we have
$$g_k(\beta) \leq b(k)^{1/k-1}, \quad \text{where } b(k) = \frac{k^k}{(k-1)^{k-1}}.$$

Routine calculation shows that
$$g_k'(\beta) = \left(\frac{\beta}{k^2} + 1 - \beta\right)^{k-1}\frac{1}{k^2}\left(1 - \frac{1}{k^2}\right)[k^2 - k + 1 - (k^3 - k)\beta].$$
It follows that $g_k(\beta)$ is maximized at
$$\beta_k = \frac{k^2 - k + 1}{k^3 - k},$$
at which point its value is
$$g_k(\beta_k) = \frac{(k^2 + 1)(k-1)^{k-1}}{k^{3k}}.$$
Therefore we need to show that for all $k \geq 2$,
$$\frac{(k^2 + 1)(k-1)^{k-1}}{k^{3k}} \leq b(k)^{-(1-1/k)} = \frac{(k-1)^{(k-1)(1-1/k)}}{k^{k-1}}.$$
Rearranging, we need to show that
$$(k^2 + 1)(k-1)^{(k-1)/k} \leq k^{2k+1}.$$
This holds since $k^2 + 1 \leq 2k^2 \leq k^3 \leq k^{2k}$ and $(k-1)^{(k-1)/k} \leq k$.

## C.3 Lower bounds for randomized learners

In this section, we prove lower bounds for randomized learners. We prove a lower bound which is tight when the adversary is adaptive for all triplets $n, k, r$. We also prove a lower bound which is tight even when the adversary is oblivious, but only for all triplets $n, k, r$ satisfying $r = O(\log_k n)$.

We first prove a near-optimal lower bound on $\mathsf{opt}^{\mathrm{obl}}_{\mathrm{bandit}}(n, k)$.

**Lemma C.10.** *Let $n \geq k \geq 2$. Then*

$$\mathsf{opt}^{\mathrm{obl}}_{\mathrm{bandit}}(n, k) \geq \frac{k-1}{2} \lfloor \log_k n \rfloor.$$

*Proof.* Assume without loss of generality that $n = k^m$ for some integer $m$ (otherwise replace $n$ with the largest power of $k$ below it). We label the set of experts using the elements of $[k]^m$, that is, for each $y_1, \ldots, y_m \in [k]$ there is an expert $h(y_1, \ldots, y_m)$. We choose the correct expert at random.

We label the instances by $x_1, \ldots, x_m$, where the prediction of $h(y_1, \ldots, y_m)$ on $x_i$ is $y_i$. The fixed sequence of instances is

$$\underbrace{x_1, \ldots, x_1}_{k \text{ times}}, \ldots, \underbrace{x_m, \ldots, x_m}_{k \text{ times}}.$$

For any randomized strategy of the learner, it will discover the correct label of $x_i$ after having made at least $\frac{k-1}{2}$ mistakes in expectation (see for example [Daniely and Helbertal, 2013, Claim 2]). Therefore the expected number of mistakes incurred by the learner is at least $\frac{k-1}{2} m$. Consequently, for each learner there is an expert $h(y_1, \ldots, y_m)$ on which it makes at least as many mistakes in expectation. $\qquad\square$

Let us now prove a lower bound for the $r$-realizable case. In contrast with the realizable case, this bound holds only against an adaptive adversary.

**Lemma C.11.** *Let $n \geq k \geq 2, r \geq 0$. Then*

$$\mathsf{opt}^{\mathrm{adap}}_{\mathrm{bandit}}(n, k, r) \geq (k-1)r/2.$$

*Proof.* For better readability assume that $r$ is even. Suppose that $n = k$. In every round, the adversary let expert $i$ predict the label $i$. We describe an adversary which chooses the correct labels at random and forces the learner to make at least $(k-1)r/2$ mistakes in expectation. Therefore, there exists an adversary which chooses the correct labels deterministically and achieving that. Before the game starts, The adversary draws a random sequence $y_1, \ldots, y_{kr/2}$ of $kr/2$ labels independently and uniformly. In round $t$, the adversary chooses $y_t$ as the correct label. The adversary ends the game when one of the following conditions hold:

1. The learner has predicted correctly in $r/2$ rounds.

2. The game was played for $kr/2$ many rounds.

First, we show that the learner makes at least the stated number of mistakes in expectation. There are two cases to consider. If the game is finished because $kr/2$ rounds have passed and the learner has not yet made $r/2$ correct predictions, then the learner has made at least $(k-1)r/2$ mistakes with probability one, concluding this case.

In the second case, we assume that the game is finished when the learner makes exactly $r/2$ correct predictions. Partition the rounds of the game into $r/2$ intervals: the first interval starts at the first round. Each interval ends when the learner makes a correct prediction. The expected number of rounds in each interval is stochastically dominated by a $\mathrm{Geom}(1/k)$ random variable. Since in every interval there is exactly a single correct prediction, the total expected number of mistakes in all intervals is $(k-1)r/2$.

It remains to show that the adversary is $r$-consistent. Assume without loss of generality that the best expert is not consistent with all correct predictions of the learner, which is at most $r/2$. Now, recall that the number of rounds in the game is at most $kr/2$. Therefore, there must be a label $i^\star$ predicted by the learner in at most $r/2$ many rounds. The best expert is at least as good as the expert

$i^\star$, which predicts $i^\star$ in all rounds. The adversary is thus $r/2$-consistent on rounds where the learner is correct, and $r/2$-consistent on rounds where the learner is incorrect. In total, the adversary is $r$-consistent. □

In the proof, we only used the adversary's adaptivity to decide on the number of rounds in the game, and not on the correct labels. Therefore, one might conjecture that it is possible to use the same technique for proving a lower bound also against an oblivious adversary. However, it is not possible; as we show in Proposition E.2, it holds that $\mathsf{opt}_{\mathrm{bandit}}^{\mathrm{obl}}(k, k, r) = \Theta(k + r)$ for all $k \geq 2, r \geq 0$. The essential reason allowing Proposition E.2 is that an oblivious adversary must be $r$-realizable in a strong sense, as discussed in Remark A.1.

### C.4  Tight bounds for randomized learners

We can now deduce tight bounds on $\mathsf{opt}_{\mathrm{bandit}}^{\mathrm{adap}}(n, k, r)$, and bounds on $\mathsf{opt}_{\mathrm{bandit}}^{\mathrm{obl}}(n, k, r)$ which are tight in the regime $r = O(\log_k n)$.

**Theorem C.12.** *Let $n \geq k \geq 2, r \geq 0$. Then*

$$\mathsf{opt}_{\mathrm{bandit}}^{\mathrm{adap}}(n, k, r) = \Theta(k(\log_k n + r)).$$

*Furthermore, if $r \leq C \cdot \log_k n$ for some constant $C$ then*

$$\mathsf{opt}_{\mathrm{bandit}}^{\mathrm{obl}}(n, k, r) = \Theta_C(k \log_k n)$$

*where $\Theta_C$ hides a constant that depends on $C$.*

*Proof.* The bound $\mathsf{opt}_{\mathrm{bandit}}^{\mathrm{adap}}(n, k, r) = \Theta((\log_k n + r)k)$ is implied by putting together the upper bound of Theorem C.7, and the lower bounds of Lemmas C.10 and C.11. The lower bound in $\mathsf{opt}_{\mathrm{bandit}}^{\mathrm{obl}}(n, k, r) = \Theta_C(k \log_k n)$ for $r \leq C \cdot \log_k n$ easily follows from Lemma C.10, and the upper bound from Theorem C.7. □

## D  The role of information

In this section, we prove the following theorem, which implies Theorem 1.1.

**Theorem D.1** (Full information vs. bandit feedback)**.** *For every pattern class $\mathcal{P} \subset (\mathcal{X} \times \mathcal{Y})^\star$ it holds that*

$$\mathsf{opt}_{\mathrm{bandit}}^{\mathrm{adap}}(\mathcal{P}) \leq 6k \cdot \mathsf{opt}_{\mathrm{full}}^{\mathrm{rand}}(\mathcal{P}).$$

Theorem D.1 will also allow us to extend Theorem 1.1 (which is formulated for concept classes) to the $r$-realizable setting.

### D.1  Reduction from bandit to full-information feedback

Given Theorem C.1, the main ingredient left in order to prove Theorem D.1 is the following reduction from learning with bandit feedback to learning with full-information feedback. Let us briefly describe the idea, inspired by [Long, 2020, Hanneke, Livni, and Moran, 2021].

The upper bound we prove on $\mathsf{opt}_{\mathrm{bandit}}^{\mathrm{adap}}(n, k, r)$ makes no assumptions on the way the experts make their predictions. Therefore, it holds in particular for the case where the experts' predictions are determined by algorithms chosen by the learner. We can exploit this property to reduce learning with bandit feedback to learning with full-information feedback. Let $A$ be a deterministic full-information learning algorithm that is guaranteed to make at most $r$ many mistakes against any adversary. Whenever $A$ is consistent with the feedback, we do nothing. When $A$ is inconsistent with the feedback, we can be sure that it has made a mistake. We thus create $k$ different copies of $A$, each guessing a different full-information feedback, exactly one of which is correct. Given that $A$ makes at most $r$ many mistakes when provided with full-information feedback, an optimal randomized algorithm for the bandit feedback model will thus make at most $\mathsf{opt}_{\mathrm{bandit}}^{\mathrm{adap}}(k^r, k, r) = O(k \cdot r)$ mistakes against an adaptive adversary. The details are made precise in the proof of the proposition below.

**Proposition D.2.** *Let $A$ be a deterministic learning algorithm for a pattern class $\mathcal{P} \subset (\mathcal{X} \times \mathcal{Y})^\star$ in the full-information setting, which makes at most $r$ many mistakes on any sequence $S \in \mathcal{P}$. Then,*

$$\mathsf{opt}^{\mathrm{adap}}_{\mathrm{bandit}}(\mathcal{P}) \leq 3k \cdot r.$$

*Proof.* We employ an optimal algorithm for prediction with expert advice, where the experts result from running $A$ and translating the bandit feedback into a full-information feedback in all possibly ways.

At each round $t$ we will maintain a $k$-ary tree $T$ of depth at most $r$, each of whose leaves is labelled by a sequence of $d \leq r$ tagged examples $(t_1, x_{t_1}, y_{t_1}), \dots, (t_d, x_{t_d}, y_{t_d})$. Each internal node has its outgoing edges labelled by the elements of $\mathcal{Y}$. Initially, $T$ consists of a single node labelled by the empty sequence. Additionally, we communicate with an optimal learner $B$ for the problem of prediction with expert advice in the $r$-realizable setting, with $k^r$ experts indexed by $[k]^r$.

At round $t$, the adversary sends an instance $x_t$ to the learner. The learner uses $A$ to generate a prediction for each leaf: if the leaf is labelled by the sequence $\sigma$, then the corresponding prediction is $A(\sigma, x_t)$. These predictions are converted to expert predictions as follows. A leaf $\ell$ at depth $d$ has an "address" $\alpha \in [k]^d$ formed by the labels of the edges on the path from the root. All experts whose index extends $\alpha$ make the same prediction as $\ell$. These expert predictions are sent to the optimal learner $B$, which provides a label $\hat{y}_t$. The label is forwarded to the adversary, which responds with either $\rightarrow$ or $\nrightarrow$.

The learner now updates the tree as follows. For each leaf in the tree, it checks whether it predicts a label which is inconsistent with the adversary's feedback. This can happen in two ways: either the leaf predicted $\hat{y}_t$ and the adversary feedback was $\nrightarrow$, or the leaf's prediction was different from $\hat{y}_t$ and the adversary feedback was $\rightarrow$. For each such leaf, if the leaf is at depth $r$, then we mark it as *bad* (a leaf not marked as *bad* is a *good leaf*). Otherwise, suppose that the leaf is labelled by $\sigma$. We add to the leaf $k$ children, and label the $y$'th child by $\sigma \circ (t, x_t, y)$. This completes the description of the algorithm.

Let $S^\star \in \mathcal{P}$ be a pattern that the adversary is consistent with at the end of the game. We say that a label $(t_1, x_{t_1}, y_{t_1}), \dots, (t_d, x_{t_d}, y_{t_d})$ is *consistent with $S^\star$* if $S^\star_{t_i} = (x_{t_i}, y_{t_i})$ for $i \in [d]$. The definition of the algorithm ensures that the following properties hold at the conclusion of each round $t$:

1. Each good leaf at depth $d$ corresponds to an expert whose predictions are inconsistent with the bandit feedback for exactly $d$ rounds.
   This is because we add children to a leaf precisely when its prediction is inconsistent with the bandit feedback (unless it is already at maximal depth, in which case we mark it as *bad*).

2. For each good leaf whose label $\sigma$ is consistent with $S^\star$, if we run $A$ on the sequence $\sigma$, then it makes a mistake at every round.
   Indeed, when we add a new leaf labelled $\sigma \circ (t, x_t, y_t)$, where $S^\star_t = (x_t, y_t)$, the prediction of $A$ on $x_t$ must have been incompatible with the bandit feedback, and in particular, with $y_t$, which is compatible with the bandit feedback.

3. There is always a good leaf whose label is consistent with $S^\star$.
   Indeed, consider any leaf $\ell$ satisfying this at time $t-1$, let its label be $\sigma$, and let $S^\star_t = (x_t, y_t)$. If $\ell$ is at depth $r$ then $A$ must output $y_t$ on input $(\sigma, x_t)$ due to the preceding item (it cannot make more than $r$ mistakes on any sub-sequence of $S^\star$). Consequently, $\ell$ doesn't become bad.
   If $\ell$ is at lower depth and its prediction is inconsistent with the bandit feedback, then its child labelled $\sigma \circ (t, x_t, y_t)$ satisfies the required properties.

In particular, due to items (1) and (3) there is an expert whose predictions are inconsistent with the bandit feedback for at most $r$ rounds. Using the algorithm given by Theorem C.7 results in a learner whose loss is at most

$$\mathsf{opt}^{\mathrm{adap}}_{\mathrm{bandit}}(k^r, k, r) \leq k \log_k(k^r) + 2kr = 3kr,$$

as desired. $\qquad\square$

## D.2 Proof of Theorem D.1

*Proof of Theorem D.1.* We use Proposition D.2 with Littlestone's [Littlestone, 1988] SOA algorithm for the class $\mathcal{P}$ as the full-information algorithm $A$. Originally, SOA was formulated in terms of concept classes rather than pattern classes, but adapting it to pattern classes is straightforward (see e.g. [Moran, Sharon, Tsubari, and Yosebashvili, 2023]). Since the adversary is consistent with $\mathcal{P}$, it is known that the SOA algorithm will make at most $2\mathsf{opt}_{\text{full}}^{\text{rand}}(\mathcal{P})$ mistakes as long as it receives full-information feedback. Proposition D.2 now provides the stated bound. $\qquad\square$

## D.3 An extension to the agnostic setting

In Section C we considered the $r$-realizable setting, in which the adversary is only $r$-consistent with the class of experts. The exact same setting can be considered when learning concept classes in general. Furthermore, in Section G we explain how to remove the assumption that $r$ is given to the learner.

Our bounds hold for all pattern classes, allowing an immediate adaptation of our results for concept classes to the $r$-realizable setting. Indeed, as mentioned also in Section C, for every concept class $\mathcal{H}$ and a budget function $B\colon \mathcal{H} \to \mathbb{R}$ we can define the pattern class appearing in (15):

$$\mathcal{P}(\mathcal{H}, B) = \left\{ p \in (\mathcal{X} \times \mathcal{Y})^\star : \left[ \exists h \in \mathcal{H} : \sum_{(x,y) \in p} \mathbb{1}[h(x) \neq y] \leq B(h) \right] \right\}.$$

Choosing $B = B_r$, where $B_r$ gives the budget $r$ to every $h \in \mathcal{H}$, simulates the class $\mathcal{H}$ in the $r$-realizable setting. To transform bounds for pattern classes in the realizable setting to bounds for concept classes in the $r$-realizable setting, we only need the following bounds on $\mathsf{opt}_{\text{full}}^{\text{det}}(\mathcal{P}(\mathcal{H}, B_r))$, which were originally proved for the binary case $k = 2$, but can be extended in a straightforward manner to arbitrary $k$.

**Theorem D.3** ([Cesa-Bianchi et al., 1996, Auer and Long, 1999, Filmus et al., 2023]). *For every concept class $\mathcal{H}$ and $r \geq 0$ it holds that*

$$\mathsf{opt}_{\text{full}}^{\text{det}}(\mathcal{P}(\mathcal{H}, B_r)) = \Theta(\mathsf{opt}_{\text{full}}^{\text{det}}(\mathcal{H}) + r).$$

The bounds obtained using Theorem D.3 are summarized in Table 1.

# E The role of adaptivity

In this section we prove the following result, which implies Theorem 1.2.

**Theorem E.1** (Oblivious vs. adaptive adversaries). *For every natural $k \geq 2$ there exists a pattern class $\mathcal{P} \subset (\mathcal{X} \times \mathcal{Y})^\star$ with $|\mathcal{Y}| = k$ and so that*

$$\mathsf{opt}_{\text{bandit}}^{\text{adap}}(\mathcal{P}) = \Omega(k \cdot \mathsf{opt}_{\text{bandit}}^{\text{obl}}(\mathcal{P})).$$

The proof employs the concept class $\mathcal{H}$ which consists of all constant functions. This corresponds to the classical bandit problem, in which there are only labels. We use the notation $\mathsf{opt}_{\text{bandit}}^{\text{obl}}(\mathcal{H}, r)$ for $\mathsf{opt}_{\text{bandit}}^{\text{obl}}(\mathcal{P}(\mathcal{H}, B_r))$, and similarly for other setups.

**Proposition E.2.** *Let $\mathcal{H} \subset \mathcal{Y}^\mathcal{X}$ be the hypothesis class consisting of all constant functions, and let $r \geq 0$. If $k = |\mathcal{Y}| \geq 2$ then*

$$\mathsf{opt}_{\text{bandit}}^{\text{obl}}(\mathcal{H}, r) = \Theta(k + r).$$

*Proof.* The lower bound follows easily from [Daniely and Helbertal, 2013, Claim 2]. For the upper bound, we describe a learner which makes at most $34(k + r)$ mistakes in expectation:

**First phase** For $16(k + r)$ rounds, predict each label with probability $1/k$. If there are at least $8(k + r)/k$ correct predictions, set $y$ to be the plurality vote among all rounds in which the learner guessed correctly, and move to the second phase. Otherwise, repeat the first phase.

**Second phase** In this phase, the learner always predicts $y$, switching back to the first phase once it makes $r + 1$ mistakes in the phase.

In order to analyze the expected number of mistakes, we assuming (without loss of generality) that the input sequence is infinite, and let $y^\star \in \mathcal{Y}$ be the hypothesis which is $r$-consistent with the input sequence chosen by the oblivious adversary.

Let us analyze the expected number of mistakes. In the first phase, we consider $16(k + r)$ rounds, and therefore the expected number of correct guesses is $\mu = \frac{16(k+r)}{k}$. Let $X$ be a random variable counting the number of correct predictions in the first phase. Since $\mu \geq 16$, and since $X$ is a sum of independent random variables taking values in $\{0, 1\}$, applying Chernoff's bound implies that

$$\Pr[X \leq 8(1 + r/k)] \leq \Pr[X \leq \mu/2] \leq e^{\frac{-\mu}{8}} \leq 1/e^2 < 1/4. \tag{17}$$

There are at most $r$ many examples which are not labeled by $y^\star$. Therefore, in expectation, there are at most $r/k$ many examples which are both correctly classified and whose label is not given by $h^\star$. Let $Y$ be a random variable counting all examples that are not labeled by $y^\star$ from the correctly classified examples in the first phase. Using Markov's inequality, we have:

$$\Pr[Y \geq X/2 \mid X \geq 8(1 + r/k)] \leq \frac{\mathbb{E}[Y]}{X/2} \leq \frac{r/k}{4r/k} = 1/4. \tag{18}$$

If $y = y^\star$ then the learner will make at most $r$ more mistakes in the second phase.

In order to finish the proof we calculate the expected number of times this two phase process will occur until that $y = y^\star$. In the worst case, by (17),(18) this number of times is dominated by a $\mathrm{Geom}\left((3/4)^2\right)$ random variable, which means that this process will occur at most $(4/3)^2 < 2$ times in expectation. Overall the expected number of mistakes is at most $2(16(k+r)+r) < 34(k+r)$. $\quad\square$

We can now prove Theorem E.1.

*Proof of Theorem E.1.* Given $k \geq 2$, we choose the pattern class $\mathcal{P} = \mathcal{P}(\mathcal{H}, B_k)$, where $\mathcal{H}$ consists of all constant functions.

Lemma C.11 shows that $\mathrm{opt}^{\mathrm{adap}}_{\mathrm{bandit}}(\mathcal{P}) = \Omega(k^2)$, while Proposition E.2 shows that $\mathrm{opt}^{\mathrm{obl}}_{\mathrm{bandit}}(\mathcal{P}) = O(k)$, yielding the stated bound. $\quad\square$

The proof of Lemma C.11 goes through with any hypothesis class of size $k$, where $k = |\mathcal{Y}|$. We conclude that the worst-case dependence on $r$ in $\mathrm{opt}^{\mathrm{obl}}_{\mathrm{bandit}}(\mathcal{H}, r)$ is related to the size of $\mathcal{H}$. For example, every $\mathcal{H}$ with $|\mathcal{H}| = k$ satisfies $\mathrm{opt}^{\mathrm{obl}}_{\mathrm{bandit}}(\mathcal{H}, r) = \Theta(k + r)$. However, there are classes $\mathcal{H}$ with $|\mathcal{H}| = k^r$ which satisfy $\mathrm{opt}^{\mathrm{obl}}_{\mathrm{bandit}}(\mathcal{H}, r) = \Omega(kr)$ due to Lemma C.10. This is in contrast to $\mathrm{opt}^{\mathrm{adap}}_{\mathrm{bandit}}(\mathcal{H}, r)$: for any class $\mathcal{H}$ (of size at least $k$) it holds that $\mathrm{opt}^{\mathrm{adap}}_{\mathrm{bandit}}(\mathcal{H}, r) = \Omega(kr)$. It would be interesting to find tight bounds on the worst case $\mathrm{opt}^{\mathrm{obl}}_{\mathrm{bandit}}(\mathcal{H}, r)$ for the intermediate cases where $|\mathcal{H}| = k^w$ for $w \in (1, r)$.

# F The role of randomness

In this section we prove the following result, which implies Theorem 1.4 by applying it with $d = k$.

**Theorem F.1** (Restatement of Theorem 1.5). *For every $d \geq 1$ and $k \geq 2$ there exists a concept class $\mathcal{H} \subset \mathcal{Y}^{\mathcal{X}}$ with $\mathcal{Y} = \{0, 1, \ldots, k\}$ such that*

   *1. $\mathrm{opt}^{\mathrm{det}}_{\mathrm{full}}(\mathcal{H}) = d + 1$.*

   *2. $\mathrm{opt}^{\mathrm{det}}_{\mathrm{bandit}}(\mathcal{H}) = \Theta(d \cdot k)$.*

   *3. $\mathrm{opt}^{\mathrm{adap}}_{\mathrm{bandit}}(\mathcal{H}) = \Theta(d + k)$.*

*Proof.* We first define the class $\mathcal{H} := \mathcal{H}(d,k)$. The domain is $\mathcal{X} = \{x_1, \ldots, x_{d \cdot k}\}$. For every $y \in \{1, \ldots, k\}$, define $\mathcal{H}_y$ to be the class of all functions that label all instances with $y$, except for a set $\mathcal{X}' \subset \mathcal{X}$ of at most $d$ many instances, which they label with 0. Let

$$\mathcal{H} = \bigcup_{y \in \{1, \ldots, k\}} \mathcal{H}_y.$$

We start with Item 1. For the upper bound, the learner always answers 0 until it makes the first mistake and receives as feedback a positive label $y$. Then it always answers $y$ and makes at most $d$ more mistakes. For the lower bound, the adversary asks on $x_1, \ldots, x_{d+1}$, and always tells the learner that it made a mistake. In the first round, the adversary determines the true label to be either $y = 1$ or $y = 2$, depending on the learner's prediction. In the other $d$ rounds, the adversary determines the true label to be either 0 or $y$, depending on the learner's predictions.

We now prove Item 2. For the upper bound, the learner tries to answer 1 until it makes $d+1$ mistakes, then does the same with 2, and so on until it gets to $k$. For the lower bound, the input sequence of instances the adversary asks on is $x_1, \ldots, x_{d \cdot k}$, and the adversary always provides negative feedback to the learner. The number of mistakes made by the learner is thus $d \cdot k$, and it remains to find an appropriate target function that realizes it. Since there are $d \cdot k$ rounds, there must be a label $y^\star \in [k]$ predicted by the learner for a set $\mathcal{X}'$ of instances of size at most $d$. We choose the target hypothesis $h^\star$ to be the hypothesis that gives the label 0 to the instances of $\mathcal{X}'$, and $y^\star$ to all other instances. The hypothesis $h^\star$ belongs to $\mathcal{H}$ by definition.

Finally, we prove Item 3. The lower bound is straightforward by using [Daniely and Helbertal, 2013, Claim 2]. We prove the upper bound by describing a randomized learner for $\mathcal{H}$ that makes at most $2(d + k)$ mistakes in expectation. The learner operates in two phases. In the first phase, for every instance for which it has not yet received positive feedback, it picks 0 with probability half, and all other labels with probability $1/(2k)$ each. Once the learner receives positive feedback which is given on a prediction different from 0, it enters the second phase, in which it will always predict this label, and will make at most $d$ more mistakes on instances labeled with 0.

It remains to upper bound the expected number of mistakes in the first phase. There are two types of mistakes:

1. Rounds where the adversary chooses the label 0. We divide these rounds into $d$ (or fewer) phases: the first one ends when the learner correctly guesses the first instance labeled 0; the second one ends when the learner correctly guesses the second instance labeled 0; and so on. Once the learner has found $d$ instances labeled 0, the remaining instances must have a positive label.
   Let $X_i$ (where $1 \le i \le d$) be the number of mistakes made in the $i$'th phase.

2. Rounds where the adversary chooses a positive label. Let $Y$ be the number of mistakes in such rounds.

Each of $X_1, \ldots, X_d$ is stochastically dominated by a $\mathrm{Geom}(1/2) - 1$ random variable (that is, a geometric random variable from which we subtract 1), and $Y$ is stochastically dominated by a $\mathrm{Geom}(1/2k) - 1$ random variable. It follows that the expected number of mistakes is at most

$$d(2-1) + (2k-1) < d + 2k.$$

Together with the at most $d$ additional mistakes in the second phase, we conclude that the learner makes at most $2(d + k)$ mistakes in expectation. $\qquad\square$

Theorem F.1 provides a negative answer to an open question posed by Daniely and Helbertal [2013], who ask whether $\mathrm{opt}_{\mathrm{bandit}}^{\mathrm{obl}}(\mathcal{H}) = \Omega(\mathrm{opt}_{\mathrm{bandit}}^{\mathrm{det}}(\mathcal{H}))$ for every class $\mathcal{H}$. For $\mathcal{H}$ defined in the proof of Theorem F.1 with $d = k$, we see that $\mathrm{opt}_{\mathrm{bandit}}^{\mathrm{obl}}(\mathcal{H}) = O\left(\sqrt{\mathrm{opt}_{\mathrm{bandit}}^{\mathrm{det}}(\mathcal{H})}\right)$. This separation is tight up to a logarithmic factor: the bounds $\mathrm{opt}_{\mathrm{bandit}}^{\mathrm{obl}}(\mathcal{H}) = \Omega\left(\frac{\mathrm{opt}_{\mathrm{bandit}}^{\mathrm{det}}(\mathcal{H})}{k \log k}\right)$ (following from the upper bound in [Auer and Long, 1999]) and $\mathrm{opt}_{\mathrm{bandit}}^{\mathrm{obl}}(\mathcal{H}) \ge \frac{k-1}{2}$ (e.g. by [Daniely and Helbertal, 2013]) imply that $\mathrm{opt}_{\mathrm{bandit}}^{\mathrm{obl}}(\mathcal{H}) = \Omega\left(\sqrt{\mathrm{opt}_{\mathrm{bandit}}^{\mathrm{det}}(\mathcal{H})}/\log \mathrm{opt}_{\mathrm{bandit}}^{\mathrm{det}}(\mathcal{H})\right)$ for every class $\mathcal{H}$.

---

ALGORITHM DT

**Input:** A concept class $\mathcal{H}$; a learner $A$ for $\mathcal{H}$ that when given $r \geq r^\star$ enjoys a mistake bound of $d_1(r + d_2)$, where $d_1, d_2 \geq 1$ does not depend on $r$.
**Initialize:** $r_M = d_2$.

**For** $t = 1, 2, \ldots$

1. If the best hypothesis was inconsistent with the feedback for at most $r_M$ many rounds:

   (a) Predict as $A$ predicts under the assumption that the best hypothesis is inconsistent with the feedback for at most $r_M$ many rounds, and given all information gathered by $A$ in previous rounds.

2. Otherwise:

   (a) Update $r_M := 2 \cdot r_M$.
   (b) Restart $A$.

---

Figure 2: The "doubling trick" algorithm DT.

Another interesting corollary of Theorem F.1 is that a significant separation between full-information and bandit feedback holds, in some cases, for deterministic learners but not for randomized learners. Indeed, as long as $d = \Omega(k)$, we have that both $\mathsf{opt}_{\mathrm{full}}^{\mathrm{det}}(\mathcal{H})$ and $\mathsf{opt}_{\mathrm{bandit}}^{\mathrm{adap}}(\mathcal{H})$ are of the order $O(d)$, while $\mathsf{opt}_{\mathrm{bandit}}^{\mathrm{det}}(\mathcal{H})$ is of order $\Omega(d \cdot k)$.

## G   Prediction without prior knowledge

When studying agnostic mistake bounds in previous sections, we assumed the $r$-realizability assumption, in which the number of inconsistencies the best hypothesis (or expert) has with the feedback, $r^\star$, is given to the learner (or some decent upper bound on it). In this section, we show how to remove this assumption, while suffering only a constant factor degradation in the mistake bound. To achieve this goal, we use a "doubling trick" in the same spirit of the doubling trick used in [Cesa-Bianchi et al., 1997, Section 4.6]. The algorithm DT, described in Figure 2, receives as input an algorithm $A$ for the $r$-realizable setting. That is, $A$ requires knowledge of a decent upper bound $r \geq r^\star$ to work well. Algorithm DT converts it to an algorithm which works well without this prior knowledge, in the following way. It operates in phases, where in the beginning of each phase it resets the memory of $A$, and guesses a bound $r_M$ on $r^\star$ which is larger than the value guessed in the previous phase. When the best expert in this phase reaches $r_M + 1$ inconsistencies, we deduce that our guess of $r_M$ was incorrect, and move on to the next phase. Thus, once $r_M \geq r^\star$, it is guaranteed that DT enters its last phase. The values we choose for $r_M$ guarantee that the obtained mistake bound of DT is not much worse than of $A$, when given the knowledge of $r^\star$.

Let $\mathsf{DT}(A)$ denote DT when executed with the $r$-realizable algorithm $A$.

**Proposition G.1.** *Let $\mathcal{H} \subset \mathcal{Y}^{\mathcal{X}}$ be a concept class. Suppose that $A$ is a bandit feedback learning algorithm for $\mathcal{H}$, that when given a bound $r \geq r^\star$, enjoys a mistake bound of $d_1(r + d_2)$, where $d_1, d_2 \geq 1$ does not depend on $r$ (but may depend on $\mathcal{H}$). Then, algorithm $\mathsf{DT}(A)$, without any prior knowledge, enjoys a mistake bound of $10d_1(r^\star + d_2)$. Furthermore, if $A$ is deterministic then so is $\mathsf{DT}(A)$.*

*Proof.* We split the execution of DT to phases according to the value of $r_M$. That is, in phase $i$, we have $r_M = 2^i d_2$, where $i \geq 0$.

We now consider two cases. In the first case, suppose that $r^\star \leq d_2$. Then the initial value $r_M = d_2$ is a correct guess, and thus DT simply runs $A$ with this assumption, and enjoys a mistake bound of $2d_1 d_2 \leq 2d_1(r^\star + d_2)$.

In the second case, let $a^\star > 0$ be such that $r^\star = 2^{a^\star} d_2$. In the worst case, the values of $r_M$ chosen by DT during its execution are $2^0 \cdot d_2, 2^1 \cdot d_2, \ldots, 2^{a-1} \cdot d_2, 2^a \cdot d_2$, where $a = \lceil a^\star \rceil$. Indeed, if $h^\star$ is the best hypothesis, then it is inconsistent with the feedback for at most $r^\star$ many times throughout the entire run, and in particular, throughout the last phase where $r_M \geq r^\star$. Since DT resets $A$ at the end of every phase, in each phase $i$, DT is guaranteed to make at most $d_1(2^i d_2 + d_2)$ mistakes in expectation. Between phases, when the best hypothesis is inconsistent with the feedback for the $(r_M + 1)$'th time, another mistake might be made by DT. By linearity of expectation, we see that the mistake bound of DT is at most

$$a + d_1 \sum_{i=0}^{a} (2^i d_2 + d_2) \leq a + d_1 \sum_{i=0}^{a} 2^{i+1} d_2 \leq 5d_1 2^a d_2.$$

Now, since $a - 1 \leq a^\star$, we have $r^\star \geq 2^{a-1} d_2$. Therefore:

$$5d_1 2^a d_2 = 10d_1 2^{a-1} d_2 \leq 10d_1 r^\star.$$

The "furthermore" part of the proposition is obvious from the definition of DT.  □

The mistake bounds proved in this work under the assumption that a bound $r \geq r^\star$ is given are typically of the form $f(|\mathcal{Y}|)(r + g(\mathsf{opt}_{\text{full}}^{\text{det}}(\mathcal{H}), |\mathcal{Y}|))$, where $f, g$ are functions to $\mathbb{R}$. Therefore, we can apply Proposition G.1 with $d_1 = f(|\mathcal{Y}|), d_2 = g(\mathsf{opt}_{\text{full}}^{\text{det}}(\mathcal{H}), |\mathcal{Y}|)$ to our bounds, and remove the assumption that a bound $r \geq r^\star$ is given.

