# OpenReview forum: "Bandit-Feedback Online Multiclass Classification: Variants and Tradeoffs"
_NeurIPS.cc/2024/Conference — NeurIPS 2024 poster_

### Official Review · Reviewer_qHYZ · 2024-07-02

**Soundness:** 3
**Presentation:** 3
**Contribution:** 3
**Rating:** 7
**Confidence:** 4

**Summary:**

The work studies multiclass classification with a finite label set $\mathcal{Y}$ and a model class $\mathcal{H}$ of $\mathcal{Y}$ -valued functions. The analysis focuses on optimal mistake bounds as opposed as optimal regret. In particular, the goal is to find tight relationships between optimal regret bounds, $\mathrm{opt}(\mathcal{H})$ for deterministic and randomized algorithms, against oblivious and randomized adversaries, in the full and bandit feedback models. The main contribution is an upper bound relating $\mathrm{opt}(\mathcal{H})$ for randomized algorithms and adaptive adversaries in the bandit model to $\mathrm{opt}(\mathcal{H})$ for randomized algorithms in the full feeedback model. The main result is proven through an auxiliary technical result proving a nearly optimal randomized mistake bound for the problem of prediction with expert advice in the $r$-realizable setting (where $r$ is the mistake bound of the best expert).

**Strengths:**

The work provides new results characterizing the price of bandit information in the optimal mistake boind for multiclass classification. The topic has been intensively studied in the literature on learning theory and is still of interest for the community. The technical contribution is original and nontrivial. The presentation is clear and the results are well placed in the context of previous works. It is obvious that the authors are very familiar with the topic. The connection with prediction with expert advice is interesting.

**Weaknesses:**

The contribution is more technical than conceptual. Many of the proof techniques are extensions of previous work, but the work is honest about it. The relationship between mistake bounds and regret bounds for the same setting could have been fleshed out better. It would be useful to mention absolute (as opposed to relative) upper and lower bounds on the various $\mathrm{opt}(\mathcal{H})$ whenever they are available.

**Questions:**

Can you elaborate more on the relationship between mistake bounds and regret bounds for bandit multiclass classification? I am also referring to the extent to which proof techniques can be re-used in the other setting.

Can you list the available bounds on the various $\mathrm{opt}(\mathcal{H})$ that only depend on the hypothesis class and the cardinality of the label class?

**Limitations:**

There is not an explicit section about limitations, but the authors adequately point out the open questions.
The work is theoretetical with no immediate societal impact.

---

> ### Author Rebuttal · Authors · 2024-08-04
>
> Response to weakness:
>
> The main goal of the paper is to measure the role of natural and well-studied  resources given to the learner/adversary, and such bounds are inherently relative. However, it is well-known that  $\mathsf{opt}_{\operatorname{full}}^{\operatorname{det}}(\mathcal{H})$ (and up to a factor of $2$ also its randomized parallel)
> is exactly quantitatively characterized by the Littlestone dimension [1,2], which is a combinatorial dimension depending only on the class $\mathcal{H}$. Therefore, the bounds of Theorem 1.1 can be seen as absolute, and we agree that we should clarify it in the paper. Further, the upper bounds in Theorem 1.2 and 1.4  are also absolute,  as witnessed by Equation (1). The absolute bounds implying the lower bound of Theorem 1.2 are explicitly written in lines 93-97. The absolute bounds implying the lower bound of Theorem 1.4 are explicitly stated and proved in Appendix F (Theorem F.1). In the next version of the paper, we will add the statement of Theorem F.1 to the main text.
>
>
> Response to question #1:
>
> The relationship between a mistake bound and its matching regret bound is “regret bound = mistake bound - $r$”, where $r$ is the number of mistakes made by the best hypothesis in class. Therefore, every mistake bound can be converted to a regret bound and vice versa, via this equation. We will add this formal equation to the paper, to make this relationship more clear.
>
>
> Response to question #2:
>
> See response to weakness.
>
>
> References:
>
> [1] Nick Littlestone. Learning quickly when irrelevant attributes abound: A new linear-threshold 390 algorithm. Machine learning, 2(4):285–318, 1988.
>
> [2] A. Daniely, S. Sabato, S. Ben-David, and S. Shalev-Shwartz. Multiclass learnability and the erm principle. In COLT, 2011.

---

> > ### Comment · Reviewer_qHYZ · 2024-08-08
> >
> > Thanks for addressing my questions.
> >
> > Your "response to weakness" is addressing very well my comment.
> >
> > Your "Response to question #1" is missing the point: I wanted to know the extent to which existing proof techniques for bounding regret may be re-used for proving your mistake bounds. I am asking this because it would help relate your results with the existing results for regret in bandit multiclass classification.

---

> > > ### Author Response · Authors · 2024-08-08
> > >
> > > Thank you for responding and clarifying.
> > >
> > > In the context of prediction with expert advice, we discuss this in detail in lines 296-313.
> > > The main message of those lines is that to the best of our knowledge, known regret bounds are proved for a different, easier in some sense (for the learner) definition of $r$-realizability than ours. Our definition only requires that the best expert is inconsistent with the adversary's feedback for at most $r$ many rounds. Known bounds use a definition requiring that the best expert makes at most $r$ many mistakes. In our definition, even the best expert might make much more than $r$ many mistakes. This definition is useful when proving Theorem 1.1, as explained in Section 2.1.
> > >
> > > However, if the adversary is oblivious, then both definitions of $r$-realizability coincide. The reason we can not simply convert a regret bound in this setting to a tight mistake bound, is that known regret bounds depend on the number of rounds in the game. Mistake bounds obtained from such regret bounds could only be tight in specific regimes where $r$ is relatively large.
> > >
> > > As for learning concept classes, our mistake bounds, as well as known regret bounds [1] are proved via a reduction to prediction with expert advice, so a similar discussion holds for this problem as well.
> > >
> > > [1] Amit Daniely and Tom Helbertal. The price of bandit information in multiclass online classification.
> > > 374 In Conference on Learning Theory, pages 93–104. PMLR, 2013.

---

### Official Review · Reviewer_WVbQ · 2024-07-12

**Soundness:** 2
**Presentation:** 2
**Contribution:** 2
**Rating:** 5
**Confidence:** 2

**Summary:**

This paper studies the mistake bounds of multiclass classification with adversaries. This paper provides the mistake bound gaps between the bandit feedback setting and full information setting, between the adaptive and oblivious adversaries for randomized learners under the bandit feedback setting, and between the randomized and deterministic learner.

**Strengths:**

- This work considers various settings.
- The proposed gaps between adaptive and oblivious settings and between randomized and deterministic learners are nearly optimal.
- Several future directions are discussed.

**Weaknesses:**

- For the agnostic setting with oblivious adversaries, the mistake bound is not tight for large $r^*$.
- The randomized algorithm proposed for the expert setting involves minimax calculation, which is inefficient.

**Questions:**

N/A

**Limitations:**

No potential negative societal impact.

---

> ### Author Rebuttal · Authors · 2024-08-04
>
> Response to weakness #1:
>
> As stated in lines 310-313, we are mainly interested in the experts setting as a means for proving Theorem 1.1. We believe that the paper already explores a sufficient range of problems and variations within the setting of learning hypothesis classes. Therefore, we intentionally leave the oblivious adversary case of the experts setting for future work, as solving it may require a fundamentally different approach.
>
> Response to weakness #2:
>
> It is indeed an interesting open question to find a more natural and efficient algorithm for the experts setting. This question is discussed in detail at the open questions section.

---

> > ### Comment · Reviewer_WVbQ · 2024-08-13
> > **Response to Rebuttal**
> >
> > Thanks for your response, and I have no further questions.

---

### Official Review · Reviewer_x7iL · 2024-07-16

**Soundness:** 4
**Presentation:** 2
**Contribution:** 4
**Rating:** 6
**Confidence:** 2

**Summary:**

This paper considers online multi class classification in the adversarial setting with a focus on the worst-case expected number of mistakes as a performance measure. The paper studies the effects of information (i.e. feedback provided to the learner), adaptivity (of the adversary) and randomness (of the learner), and derives new upper and lower bounds relevant to each of these considerations.

For information, the paper solves an open problem from Daniely and Helbental (2013) and provides results for showing the price of bandit versus full-information for randomised learners. The proof of this theorem is the main technical contribution of the paper and relies on a reduction of the problem to a particular instance of prediction with expert advice, and the derivation of new bounds for the prediction with expert advice problem. The results give upper and lower bounds on the mistakes which are tight to logarithmic factors.

Similarly, near-matching bounds are derived showing the cost of facing an adaptive adversary, rather than an oblivious one and for following a deterministic policy rather than a randomised one.

**Strengths:**

The paper answers a number of open questions in the area, providing a comprehensive piece of work which covers several important aspects of the important problem of online classification.

The theoretical contributions are non-trivial and well explained, particularly the sketches of the key ideas of Theorem 1.1 and 1.5 in section 2, which highlight the main novel contributions of the paper.

The connections to existing literature are very well established within the text.

**Weaknesses:**

The main weakness of the paper is its readability. A lot of key concepts are deferred to the appendices, presumably to highlight the most technically impressive contributions sooner, and I feel that the paper would be more accessible if more space were given to describing fundamental aspects of the problem in the introductory sections. See questions section for some specific suggestions, but a general commitment to making the paper easier to understand would be welcome.

**Questions:**

Lines 21-29 would facilitate easier understanding for the subsequent sections if a more explicit detail of the problem setup could be given here. Perhaps some of the definition from lines 434-442 could be ported to here, or some additional details and a reference to Appendix A?

Lines 66-69 describe the concepts of inconsistency and realisability only very briefly, in a way that is likely to be challenging for the unacquainted reader. Can you perhaps add some further details here?

Line 153: the list of related literature preceding this is dense and it is hard to parse which papers contribute what from it. It is then not absolutely clear what the precise novelty is at this point: is this the first paper to derive any bounds for multi-class prediction with expert advice and bandit feedback?

Line 135: maybe these new results should also be highlighted in the abstract if they are of independent interest?

**Limitations:**

Yes

---

> ### Author Rebuttal · Authors · 2024-08-04
>
> Contribution of Theorem 1.1:
>
> We first want to make sure that the contribution of Theorem 1.1 is completely clear.
> In the summary, the following statement is made with respect to Theorem 1.1: “The results give upper and lower bounds on the mistakes which are tight to logarithmic factors.”
> However, Theorem 1.1 establishes a tight upper bound on the role of information. This is especially important because proving a bound which is tight up to a logarithmic factor is possible even without using randomness, as stated in Equation (1). It is also known that shaving this extra log factor is impossible if the learner is deterministic, as stated in Equation (2). The open question that we solve is, whether this extra log factor can be shaved by leveraging randomness in the prediction method. Theorem 1.1 answers this question affirmatively.
>
>
> Response to Weakness:
>
> We agree that some important concepts and discussions appear only in the appendix. Since our paper studies various problems and variations, this is unfortunately unavoidable due to space constraints. We sincerely tried to keep the most important discussions in the main paper. However, we completely agree with your valuable suggestions written in the questions section. We will incorporate those in the next version of this paper, and will also make another pass to see if there are any other improvements that can be made to improve the paper’s readability. We thank you for helping us to improve the paper!
>
> Response to question/suggestion #3:
>
> Many previous works studied regret bounds for prediction with expert advice with bandit feedback. However, converting those regret bounds to mistake bounds (in which we are interested in this work) results in bounds which are generally extremely sub-optimal, as they depend on the number of rounds in the game. More details can be found in lines 306-310.

---

> > ### Comment · Reviewer_x7iL · 2024-08-10
> > **Response to Rebuttal**
> >
> > Thank you for your response and clarifications. I am inclined to retain my score and confidence score for the paper, as it remains the case that I am not the most familiar with the setting it studies (so am not fully versed on its significance and some technical details) but what I have been able to verify appears accurate and substantive and will be suitably well explained with the promised clarifications.

---

### Official Review · Reviewer_qJc1 · 2024-07-19

**Soundness:** 3
**Presentation:** 3
**Contribution:** 3
**Rating:** 7
**Confidence:** 2

**Summary:**

This paper studies online multi-class classification in the mistake bound model. It focuses on understanding how various resources affect the optimal mistake bounds of the learner. These resources concern feedback models (bandit feedback vs full information), adversarial models (adaptive vs oblivious), and learning strategies (randomized vs deterministic). The paper provides a collection of nearly tight upper and lower bounds and addresses some open problems. To prove one of the results, the paper also presents new results for the problem of prediction with expert advice under bandit feedback.

**Strengths:**

- The paper addresses several important questions in online learning. Some of these questions have previously been studied for different settings, and this paper fills some of the gaps in the literature. The paper presents an interesting collection of nearly tight upper and lower bounds that are relevant to the learning theory community. In particular, it shows that in the bandit feedback setting, adaptivity of the adversary and randomness of the learner play a bigger role in the mistake bound, in comparison with the full feedback setting.

- The results are both clean and thorough, and the paper generalizes some of them to the agnostic setting.

- The new techniques for prediction with expert advice under bandit feedback can be interesting on its own.

- The results lead to some interesting open questions and future directions.

**Weaknesses:**

- There are still some gaps in the provided upper and lower bounds. For example, in Theorem 1.2, the lower bound only applies to certain hard pattern classes. Also, the algorithm proposed for the prediction with expert advice problem does not seem practical.

- The paper is highly technical and may be challenging for non-expert readers. Many important discussion are also deferred to the appendix.

- The results are interesting from a learning-theoretic perspective; however, it would be nice if the paper can discuss the practical implications of the findings.

**Questions:**

- Are there any unique challenges in the multi-class setting as opposed to binary classification?

- Can you further discuss the implication of the lower bound in Theorem 1.4?

**Limitations:**

The settings are properly discussed.

---

> ### Author Rebuttal · Authors · 2024-08-04
>
> Response to weakness #1:
>
> It is indeed an interesting open question to find a more natural and efficient algorithm for the experts setting. This question is discussed in detail at the open questions section.
>
>
> Response to weakness #2:
>
> We agree that some important concepts and discussions appear only in the appendix. Since our paper studies various problems and variations, this is unfortunately unavoidable due to space constraints. We sincerely tried to keep the most important discussions in the main paper. In the next version of this paper, we will incorporate the suggestions of reviewer x7iL, and will make another pass to see if there are any other improvements that can be made to improve the paper’s readability.
>
>
> Response to question #1:
>
> Yes. Even with full-information feedback, multiclass learnability was only fully characterized recently, in [1]. Also, in binary classification there is no difference between bandit and full-information feedback, so the challenging nature of bandit-feedback is not manifested in binary classification.
>
>
> Response to question #2:
>
> The implications of the lower bound in Theorem 1.4 are explained in detail in lines 121-134. The significant consequence of this bound is the fact that for certain classes, while exactly quantifying the deterministic mistake bound, the bandit-Littlestone dimension can be quadratic in the randomized mistake bound. On the other hand, there are some classes for which this dimension quantifies both deterministic and randomized mistake bounds. The conclusion is that a combinatorial characterization of randomized learners (which are very common in online learning), if exists, cannot be obtained solely by the bandit-Littlestone dimension.
>
> References:
>
> [1] Steve Hanneke, Shay Moran, Vinod Raman, Unique Subedi, and Ambuj Tewari. Multiclass online learning and uniform convergence. Proceedings of the 36th Annual Conference on Learning Theory (COLT), 2023.

---

> > ### Comment · Reviewer_qJc1 · 2024-08-12
> >
> > Thank you for the responses. I intend to maintain my score. The results seem interesting, but it would be really nice if the paper can be made more readable.

---

### Author Rebuttal · Authors · 2024-08-04

We thank the reviewers for taking the time to carefully read our work, and for their thoughtful comments and suggestions to improve it. We will make our best efforts to incorporate their valuable suggestions in the next version of this paper. We respond to specific issues raised by each reviewer in a comment to each review.

---

### Decision · Program_Chairs · 2024-09-25

**Decision:**

Accept (poster)

**Comment:**

This paper makes solid contribution. Please improve clarity by incorporating reviewer's comments.